# Shadow-wall lithography of ballistic superconductor–semiconductor quantum devices

Sebastian Heedt [1,2,6✉], Marina Quintero-Pérez[2,6], Francesco Borsoi [1,6], Alexandra Fursina[2], Nick van Loo[1], Grzegorz P. Mazur [1], Michał P. Nowak [3], Mark Ammerlaan[1], Kongyi Li[1], Svetlana Korneychuk[1], Jie Shen[1], May An Y. van de Poll[1], Ghada Badawy [4], Sasa Gazibegovic[4], Nick de Jong[1,5], Pavel Aseev[2], Kevin van Hoogdalem[2], Erik P. A. M. Bakkers [4] & Leo P. Kouwenhoven[1,2]

The realization of hybrid superconductor–semiconductor quantum devices, in particular a topological qubit, calls for advanced techniques to readily and reproducibly engineer induced superconductivity in semiconductor nanowires. Here, we introduce an on-chip fabrication paradigm based on shadow walls that offers substantial advances in device quality and reproducibility. It allows for the implementation of hybrid quantum devices and ultimately topological qubits while eliminating fabrication steps such as lithography and etching. This is critical to preserve the integrity and homogeneity of the fragile hybrid interfaces. The approach simplifies the reproducible fabrication of devices with a hard induced superconducting gap and ballistic normal-/superconductor junctions. Large gate-tunable supercurrents and high-order multiple Andreev reflections manifest the exceptional coherence of the resulting nanowire Josephson junctions. Our approach enables the realization of 3-terminal devices, where zero-bias conductance peaks emerge in a magnetic field concurrently at both boundaries of the one-dimensional hybrids.

[1] QuTech and Kavli Institute of Nanoscience, Delft University of Technology, Delft, The Netherlands. [2] Microsoft Quantum Lab Delft, Delft, The Netherlands. [3] AGH University of Science and Technology, Academic Centre for Materials and Nanotechnology, Krakow, Poland. [4] Department of Applied Physics, Eindhoven University of Technology, Eindhoven, The Netherlands. [5] Netherlands Organisation for Applied Scientific Research (TNO), Delft, The Netherlands. [6] These authors contributed equally: Sebastian Heedt, Marina Quintero-Pérez, Francesco Borsoi. ✉email: Sebastian.Heedt@Microsoft.com

Hybrid superconducting/semiconducting nanowires are a promising material platform for the formation of one-dimensional topological superconductors bounded by pairs of Majorana modes[1–3]. Owing to their non-Abelian exchange statistics, these localised Majorana bound states (MBS) are the fundamental constituents for fault-tolerant topological quantum computing[4,5]. Individual qubits comprise at least four MBS in several interconnected nanowire segments with a hard induced superconducting gap[6,7]. Residual fermionic states within the gap would compromise the topological protection of the Majorana modes. Hence, a fundamental challenge in the development of topological qubits is the engineering of complex, interconnected hybrid devices with hard superconducting gaps and clean, homogeneous interfaces[8,9].

Here, we introduce a fabrication technique that resolves these challenges and provides high-quality hybrid quantum devices, reflected by the absence of chemical intermixing, a high interface transparency and hard induced gaps, while involving minimal nanofabrication steps compared with previously established methods[10,11]. Our approach is based on the deposition of superconducting thin films at a shallow angle onto semiconducting nanowires, which have been selectively placed on substrates with pre-patterned gates and shadow-wall structures. It enables complex hybrid devices while eliminating lithography, etching, and other fabrication steps after the deposition of the superconductor, in the following referred to as post-interface fabrication. While shadow-wall lithography is compatible with a large variety of materials, we utilise InSb nanowires coated with Al half-shells to induce superconducting correlations – a suitable material combination to study Majorana physics[11,12]. The homogeneity of the interface between InSb and Al ultimately determines the device quality, but it is known to have very limited chemical and thermal stability[9,13]. Therefore, the reduction or elimination of post-interface fabrication steps represents a paradigm shift that enables pristine hybrid interfaces. Similar advances in quality and reproducibility (Supplementary Note 1) were made possible by the reverse fabrication process established for carbon nanotube devices[14].

In this article, we investigate the transport properties of hybrid nanowire shadow-wall devices. Initially, we examine Josephson junctions and detect subharmonic gap features that arise from multiple Andreev reflections[15]. These junctions exhibit gate-tunable supercurrents of up to 90 nA, which is exceptionally large for InSb/Al nanowires compared to previous works on InSb Josephson junction devices[9,16,17]. The shadow-wall method also facilitates 3-terminal hybrid devices with two normal metal/superconductor (N–S) interfaces, which are crucial to corroborate earlier Majorana signatures[18–20]. We investigate the transport at a single N–S interface and observe a crossover between a hard induced gap and pronounced Andreev enhancement upon increasing the junction transparency, consistent with the expected behaviour for ballistic junctions[21,22]. Finally, we report the emergence of discrete subgap states in the tunnelling conductance at both nanowire ends and detect stable zero-energy conductance peaks that coexist at certain magnetic fields and chemical potentials.

Our fabrication method paves the way for more advanced nanowire devices, including qubit implementations[6,7,23] and other multi-terminal devices that are essential for fundamental research on topological superconductors[18,24]. The versatility of the shadow-wall technique introduces a convenient and quick way to implement new device geometries with various combinations of semiconductor and superconductor materials.

## Results

### Shadow-wall lithography

A well-established approach to realise hybrid devices is based on the epitaxial growth of nanowires followed by the in-situ evaporation of a superconductor[10,25]. This method requires a subsequent etching step to expose gate-tunable wire segments without metal. Nanowires have also been grown on opposite crystal facets of etched trenches[11,26], which enables the formation of shadowed junctions without the need to etch the superconductor[11]. The native oxide that forms during the ex-situ processing is removed prior to the deposition of the superconductor. Another recent study employed growth chips with bridges and trenches that act as selectively shadowing objects during the evaporation of a superconductor[27]. Common to those methods is that the hybrid nanowires are removed from the growth substrate following the evaporation and undergo several post-interface fabrication steps such as alignment via scanning electron microscopy (SEM), electron-beam lithography involving resist coating, or etching. The latter, in particular, degrades the electrical device performance compared with shadowed junctions[13]. Moreover, hybrid devices are prone to degradation. High-temperature processing (e.g. certain dielectric deposition methods or resist baking) cannot be performed, as it would lead to chemical intermixing at the super-/semiconductor interface[28,29]. The limited chemical stability of the interface requires sample storage in vacuum at a temperature $T < 0\,°C$, which is hardly compatible with standard fabrication methods. The low thermal budget and the additional fabrication steps limit the achievable device performance in terms of electrical noise, lithographical alignment accuracy, contamination and disorder. The considerable variation from device to device imposes singular rather than standardised designs and results in a limited reproducibility of basic transport measurements.

In contrast, the core principle of our approach is to minimise or eliminate post-interface fabrication. We have engineered scalable substrates that comprise all desired functionalities without being subject to any fabrication restrictions (e.g. thermal budget limitations) since the semiconductor nanowires are only introduced right before the superconductor deposition. As depicted in Fig. 1a, we transfer InSb nanowires[30] to these substrates onto pre-patterned bottom gates covered by a continuous dielectric layer in the vicinity of shadow-wall structures. The nanowires are loaded into a customised evaporation chamber where the native oxide is removed at $T = 550\,K$ by exposure to a directed flow of atomic hydrogen radicals. Without breaking the vacuum, Al is subsequently deposited onto the samples at $T = 140\,K$. The superconductor is evaporated at a shallow angle of 30° with respect to the substrate plane, which creates a 3-facet nanowire shell that is connected to the leads and bond pads on the substrate (Fig. 1d). As illustrated in Fig. 1b, the shadow walls enable selective deposition on both the nanowires and the substrate. Adding gaps at critical locations along the shadow walls (Fig. 1c) ensures that the leads are electrically isolated from one another while eliminating the need for post-interface fabrication such as lift-off patterning or Al etching. Figure 2a shows an exemplary device without local gates that is directly bonded to a printed circuit board for low-temperature transport measurements. Here, the p$^+$-doped Si substrate enables back-gate control of the electron density in the nanowire (see Fig. 2b).

### Materials analysis

The quality of the InSb nanowires, Al thin films, and InSb/Al interfaces is assessed by transmission electron microscopy (TEM) of cross-sectional lamellae prepared via focused ion beam (FIB). These lamellae are cut out from devices like the one depicted in Fig. 2a (cf. dashed line). A continuous high-quality polycrystalline Al layer is formed on three facets of the InSb nanowires and the samples exhibit a sharp superconductor–semiconductor interface (see Fig. 2c, e and

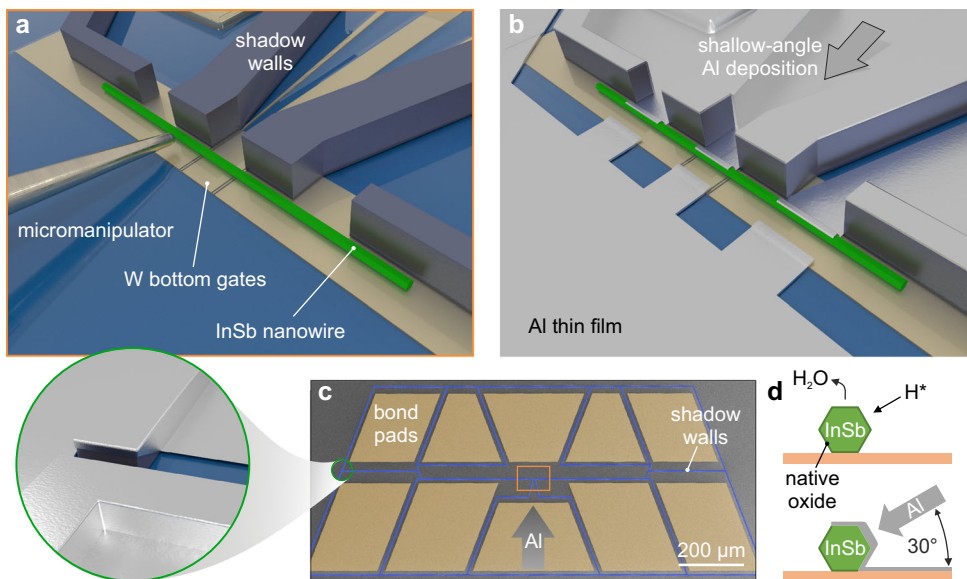

**Fig. 1 Illustration of the shadow-wall technique. a** Micromechanical transfer of the nanowires onto local bottom gates (covered by $Al_2O_3$ dielectric) in the proximity of the $Si_3N_4$ shadow walls. **b** Illustration of a final device following the H radical cleaning and Al deposition at a shallow angle. **c** False-colour SEM image of an exemplary sample prior to Al deposition. Shadow walls are designated in blue and bond pads, which are enclosed by the shadow walls, are shaded in dark yellow. Gaps are placed at critical locations along the shadow walls (cf. green circle and the illustration in the blow-up following Al deposition). This ensures that bond pads with leads are isolated from each other after the Al deposition. The area illustrated in panel **a** is indicated by the orange box. **d** Schematic of the InSb nanowire cross-section during H radical cleaning (top). The native oxide of the semiconductor is denoted by a dark green layer. The Al thin film deposited at a shallow angle of 30° forms an electrical connection from the nanowire to the substrate (bottom).

Supplementary Fig. 1). No oxide formation is observed between the Al grains, which is evident in the elemental energy-dispersive X-ray spectroscopy (EDX) composite image (Fig. 2c). The middle facet has twice the Al layer thickness (16 nm) compared to the top and bottom facets (8 nm) due to the evaporation angle of 30° with respect to the substrate plane. The InSb/Al interface is clean and there is no residual native oxide (see Fig. 2d, e), which confirms that our procedure of atomic hydrogen radical cleaning can effectively remove the oxide without damaging the InSb crystal structure. The nanowires are single-crystalline, defect-free, and exhibit a hexagonal geometry. The polycrystalline Al layer forms a continuous metallic connection from the nanowire to the substrate. This connection is crucial for the contact between the shell and the thin Al lead on the substrate and it is fundamental for more complex devices such as superconducting interferometers (see Supplementary Fig. 30) and 3-terminal Majorana devices that can reveal the opening of a topological gap[18].

**Highly transparent Josephson junctions**. We employ mesoscopic InSb/Al Josephson junctions like the one depicted in Fig. 2a to study the induced superconductivity in the nanowires. Each device comprises two Al contacts (1.8 μm wide) separated by a 110–150-nm-long bare nanowire segment that is tunable by the back-gate voltage, $V_{BG}$. The source–drain voltage, $V_{SD}$, is applied or measured between the two Al electrodes (Fig. 2b). Figure 3a shows the differential resistance, $R = dV_{SD}/dI_{SD}$, as a function of bias current, $I_{SD}$, and temperature for a typical device. The blue region ($R = 0\ \Omega$) denotes the superconducting phase, which persists up to ~1.8 K, consistent with the enhanced superconducting critical temperature for thin films with respect to bulk Al[31]. At low temperatures ($T < 0.6$ K), the hysteretic behaviour of the asymmetric $V_{SD}$–$I_{SD}$ traces is caused by self-heating of the junction. This effect disappears at higher temperatures ($T > 0.6$ K), which can be attributed to enhanced thermalisation via electron–phonon coupling[32]. Remarkably, at $T = 30$ mK, the switching current, $I_{sw}$, i.e. the observable

supercurrent, ranges from 30 to 90 nA across all devices in the open-channel regime. The magnitude of the intrinsic supercurrent, $I_c$, in ballistic and short junctions can be predicted via the Ambegaokar–Baratoff formula: $I_cR_N = \pi\Delta_{ind}/2e$, with the normal-state resistance $R_N$, the induced gap $\Delta_{ind}$, and the electron charge $e$[33]. Here, the typical $I_{sw}R_N$ product is ~110 μV, i.e. only one-third of $\pi\Delta_{ind}/2e \sim 360$ μV. The discrepancy between $I_{sw}$ and $I_c$ is consistent with previous experiments[16,17,34] and can be explained by premature switchings due to thermal activation and current fluctuations[35,36]. We note that the magnitude of $I_{sw}$ as well as the normalised quantity $eI_{sw}R_N/\Delta_{ind} \sim 0.5$ are significantly larger than in previous reports on InSb Josephson junctions[9,16,17].

In Fig. 3b, we show the differential conductance, $G = dI_{SD}/dV_{SD}$, as a function of $V_{SD}$ (red curves) for the same Josephson junction (top) and for a second device (bottom). The traces display subharmonic conductance peaks originating from multiple Andreev reflection (MAR) processes[15]. By fitting the conductance with a coherent scattering model (green curves), we can estimate the induced superconducting gap, $\Delta_{ind}$ (235 μeV and 229 μeV for device 1 and 2, respectively), and the gate-tunable tunnelling probability of the different subbands (see Supplementary Figs. 8–10)[37].

In Fig. 3c, we report the evolution of the MAR pattern as a function of magnetic field, $B_\parallel$, parallel to the nanowire axis of device 2. Here, the presence of subgap states close to the gap edge alters the typical MAR pattern and gives rise to an intricate energy dispersion in magnetic field that is further discussed in Supplementary Note 3. Eventually, the magnetic field quenches the superconductivity at a critical value of $B_c = 1.2$–1.3 T. This limit can be enhanced to about 2 T by using a thinner Al shell (Supplementary Fig. 14). These values are well above the magnetic field at which a topological phase transition should occur in hybrid InSb/Al nanowires[38].

In Fig. 3c, the out-of-gap conductance displays a dense pattern of faint peaks with an average spacing of about 30 μV and an effective Landé $g$ factor of ~20 (extracted from the energy

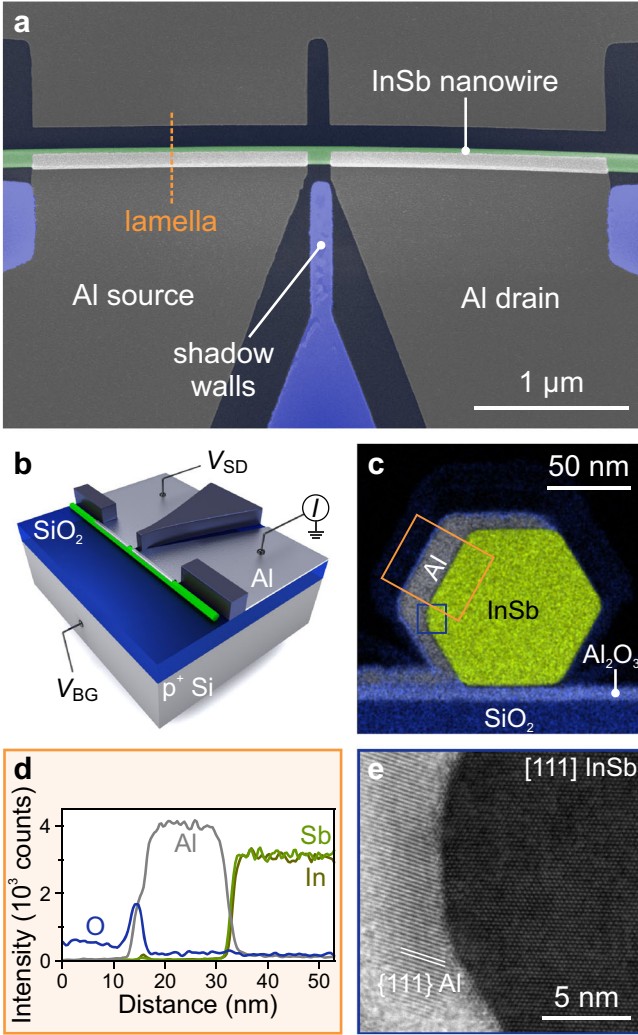

**Fig. 2 TEM analysis of the InSb/Al interface. a** False-colour SEM image of an InSb nanowire Josephson junction. **b** Schematic of the measurement setup. The back-gate voltage, $V_{BG}$, is applied to the p$^+$-doped Si substrate to tune the electron density in the nanowire. **c** Cross-sectional EDX elemental composite image of the [111] InSb nanowire covered with the Al layer and a protective layer of SiN$_x$. **d** Line-cuts of the integrated elemental counts within the orange box in panel **c**. **e** High-resolution bright-field scanning TEM image of the InSb/Al interface at the location indicated by the blue box in panel **c**.

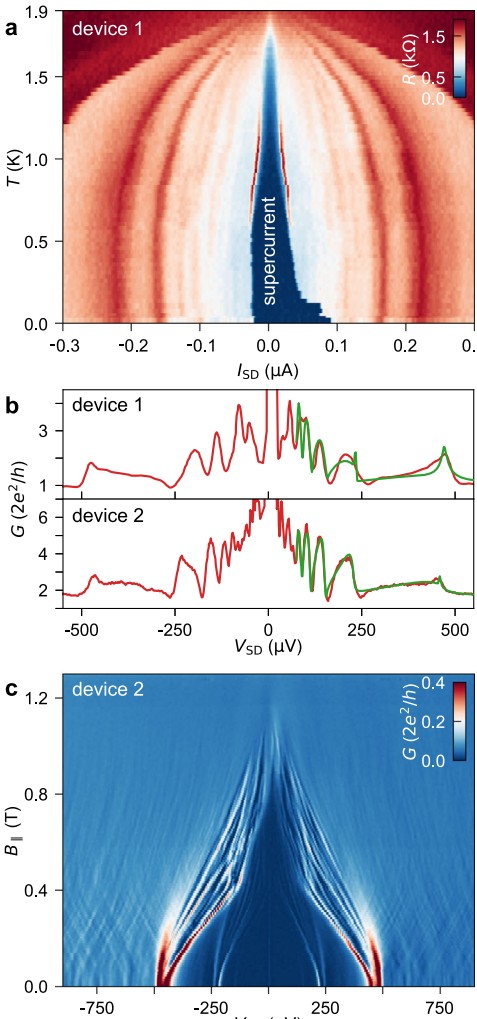

**Fig. 3 Multiple Andreev reflections and supercurrent in InSb/Al Josephson junctions. a** Differential resistance, $R$, as a function of $I_{SD}$ (upward sweep direction) and $T$ for device 1 at $V_{BG} = 13.65$ V. The switching current reaches a maximum of ∼ 90 nA at $T = 30$ mK and persists up to 1.8 K. The peaks at $I_{SD} > I_{sw}$ arise from quasiparticle transport via multiple Andreev reflections. **b** Conductance line traces (red) versus source–drain voltage for device 1 at $V_{BG} = 5.1$ V (top) and for device 2 at $V_{BG} = 3.0$ V (bottom). The theoretical fits (green) yield the transmissions, $T_n$, of the one-dimensional subbands with index $n$: $T_1 = 0.91$, $T_2 = 0.17$ (top) and $T_1 = 0.93$, $T_2 = 0.71$, $T_3 = 0.01$ (bottom). **c** Differential conductance, $G$, as a function of $V_{SD}$ and magnetic field, $B_{\parallel}$, which is oriented along the nanowire, for device 2 at $V_{BG} = -0.9$ V.

dispersion in magnetic field). This $g$ factor is larger than in Al ($|g|$ = 2) but smaller than in InSb ($|g|$ = 30–50), which indicates that these peaks stem from discrete states of the nanowire hybridised with the ones in the metal[39]. The observation of this structure might be correlated with our choice of nanowire surface treatment. In fact, the gentle atomic hydrogen cleaning preserves the pristine semiconductor crystal quality, unlike the invasive chemical or physical etching methods adopted in previous works[9,16,17,34,40].

**Hard induced gap and ballistic superconductivity.** A common technique to search for evidence of Majorana bound states is N–S tunnelling spectroscopy, which probes the local density of states. Signatures of MBS in proximitized InSb nanowires are zero-bias peaks (ZBPs) in the differential conductance at moderately large magnetic fields[41]. The ZBP height in the zero-temperature limit is predicted to be $G_0 = 2e^2/h$, independent of the tunnel-coupling strength, due to resonant Andreev reflection via a Majorana zero

mode[42]. ZBPs of non-topological origin, which mimic the subgap behaviour of MBS, may arise from disorder or potential inhomogeneities[43]. A major challenge is to reduce the detrimental role of disorder at the superconductor–semiconductor interface, which determines the final device quality. The measure of success is a hard induced gap at a finite magnetic field and quantised Andreev enhancement as a signature of ballistic transport[44,45].

An exemplary N–S device is depicted in Fig. 4a. Here, the N contact to the InSb nanowire was formed in a post-interface fabrication step, similar to the contacting of conventional shadow junctions (Supplementary Note 1). Alternatively, Al leads that are defined by the shadow walls – microns away from the N–S junction – can serve as N contacts but require additional bottom gates to render all nanowire segments fully conducting (cf. Fig. 1b). Another option to fabricate N contacts in situ involves

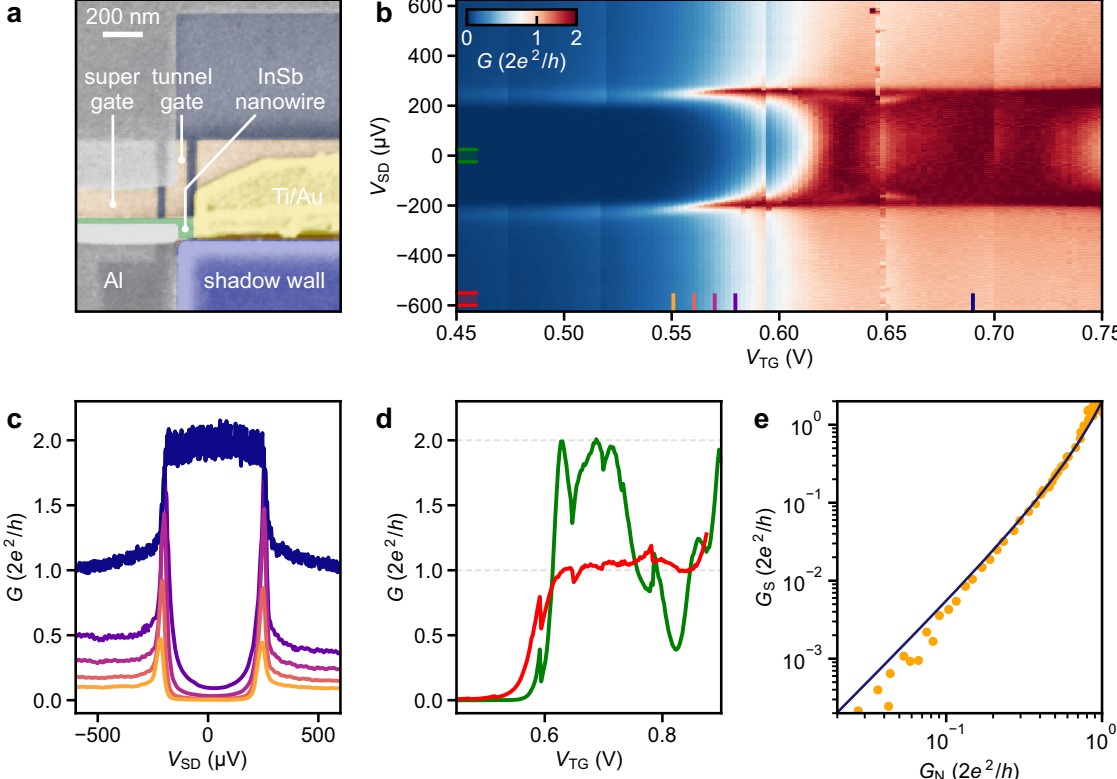

**Fig. 4 Ballistic Andreev transport. a** False-colour SEM image of an exemplary N–S junction. The W bottom gates (brown) underneath the 100-nm-wide InSb nanowire (green) are covered by 18 nm of $Al_2O_3$ dielectric. **b** Differential conductance, $G$, as a function of source–drain voltage, $V_{SD}$, and bottom tunnel-gate voltage, $V_{TG}$. The so-called super gate, which controls the chemical potential of the hybrid nanowire segment, is grounded. **c** $G$ versus $V_{SD}$ line-cuts of the data in panel **b** at the locations designated by the coloured lines. **d** Subgap conductance (green) and above-gap conductance (red) averaged over the $V_{SD}$ intervals designated in panel **b**. **e** $G_S$ (subgap conductance at zero bias) as a function of $G_N$ (normal-state conductance at $V_{SD} = 650\,\mu V$) together with the theoretically predicted dependence, which assumes Andreev-dominated transport in a single channel (blue line trace).

using two deposition angles, which we describe in detail elsewhere[46]. In Fig. 4b, we present voltage-bias spectroscopy of the N–S junction in Fig. 4a where the transmission is tunable via a pre-fabricated bottom tunnel gate. The line-cuts in Fig. 4c at low tunnel-gate voltage, $V_{TG}$, highlight the pronounced suppression of the subgap conductance, $G_S$, by about two orders of magnitude compared with the normal-state conductance, $G_N$ (cf. Supplementary Fig. 23). As the first one-dimensional subband starts to conduct fully at $V_{TG} > 0.6$ V, the above-gap conductance reaches the conductance quantum, $2e^2/h$, and the quantisation manifests itself as a plateau in the tunnel-gate dependence (Fig. 4d). At the same time, the conductance below the gap edge reaches $4e^2/h$ owing to two-particle transport via Andreev reflection[21]. This pronounced doubling of the normal-state conductance together with the quantisation of $G_N$ signifies a very low disorder strength in the junction and a strong coupling at the nanowire/Al interface[47]. While the subgap conductance reaches up to $2G_0$, it drops again at $V_{TG} \sim 0.8$ V, possibly due to inter-subband scattering as a result of residual disorder[44,47–49]. The plot of $G_S$ versus $G_N$ (Fig. 4e) follows the Beenakker model[22] reasonably well without any fitting parameter, which shows that in the single-subband regime electrical transport below the gap edge is dominated by Andreev processes. The data are well-described by the BTK theory[21] across the entire gate voltage range, demonstrating a hard induced gap of $\Delta_{ind} \sim 230\,\mu eV$ (see Methods and Supplementary Fig. 25). Discrete subgap states and ZBPs appear at a finite magnetic field and field-dependent voltage-bias spectroscopy for this N–S device is presented in Supplementary Fig. 26.

**Emergence of zero-bias peaks at both nanowire ends**. The shadow-wall technique enables 3-terminal Majorana devices for nonlocal correlation experiments[18,19] by harnessing the continuous connection of the Al shell to the substrate, as depicted in Fig. 5a. Here, the Al thin film serves as the superconducting drain lead. Established fabrication methods do not allow for the implementation of such devices since etching away the superconductor causes disorder at the InSb surface and contacting the Al shell requires selective removal of the native oxide of Al, which affects the integrity of the thin film. As shown in Fig. 5a, optional Ti/Au contacts are again added at both nanowire ends in the same fabrication run and on the same substrate as the sample in Fig. 4. With this device type, we can study the simultaneous emergence of ZBPs at both N–S boundaries in a magnetic field oriented along the wire. Here, the hybrid nanowire segment is 1 μm long and the chemical potential, $\mu$, is controlled via a bottom gate (super gate) at potential $V_{SG}$. The differential conductance is measured concurrently at both N–S boundaries by alternating the $V_{SD}$ sweep between the left and right N terminals for every increment of $B_{\parallel}$ or $V_{SG}$. Using this technique, we demonstrate the formation of zero-energy subgap states at both nanowire ends for $V_{SG} = 0$ V (see Fig. 5e, f). The effective $g$ factor extracted from the linear energy dispersion at the two boundaries is ~10, albeit the values of $g$ can be strongly gate-dependent[12]. Many experiments have demonstrated ZBPs in tunnelling spectroscopy at a single N–S boundary, indicating the presence of a robust state at zero energy[41,49–51]. The robustness of ZBPs in the parameter space (defined by chemical potential and magnetic field) has been used to substantiate their topological origin[52]. So far, no experiment

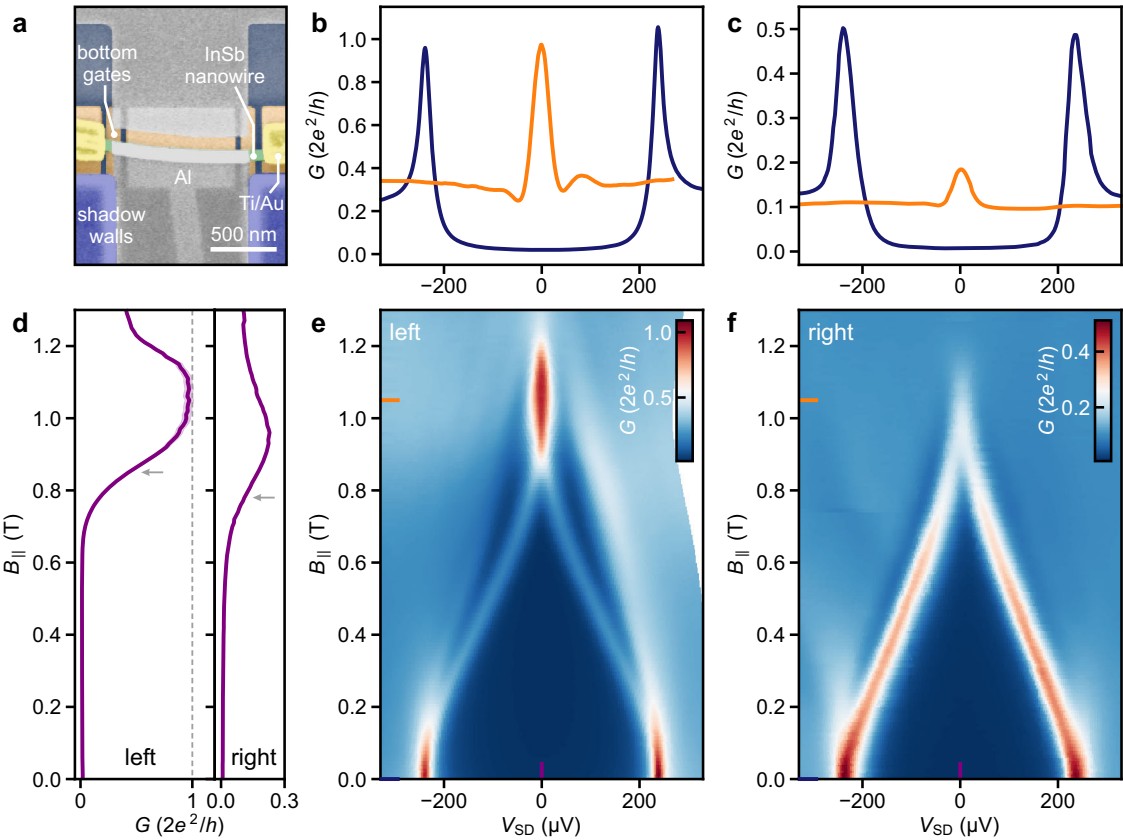

**Fig. 5 Zero-bias conductance peaks at two opposite N–S boundaries. a** False-colour SEM image of the correlation device based on an 80-nm-wide InSb nanowire with a 1-μm-long hybrid segment. **b, c** Line-cuts of the differential conductance at zero field (blue) and at $B_\| = 1.05$ T (orange) taken from panel **e** and **f**, respectively. **d** $G$ versus $B_\|$ line-cut at $V_{SD} = 0$ μV taken from panel **e** (left) and **f** (right). Shaded areas (light purple) illustrate the variation in conductance assuming an uncertainty of ±0.5 kΩ in the series resistance. For the line-cut at the right N–S junction, this variation is less than the line width. **e, f** Differential conductance, $G = dI_{SD}/dV_{SD}$, as a function of bias voltage, $V_{SD}$, and magnetic field, $B_\|$, measured concurrently at the left and right junction, respectively. Here, the super gate underneath the hybrid nanowire segment is grounded ($V_{SG} = 0$ V).

has revealed the emergence of ZBPs concurrently at both boundaries of a long hybrid nanowire. Recent experimental studies reported correlations between bound states at both ends of short (up to 400 nm long) hybrid nanowire devices[53,54]. ZBPs often originate from trivial Andreev bound states (ABS). In topological nanowires, ABS can form by overlapping MBS due to local variations in the chemical potential or random disorder, which emphasises the need for long and pristine hybrids[43]. A topological phase with well-separated MBS requires that potential inhomogeneities along the hybrid segment, $\Delta\mu$, are much smaller than the width of the topological phase, $2\sqrt{E_Z^2 - \Delta_{ind}^2}$, where $E_Z$ is the Zeeman energy[1,55]. We see in Fig. 5e,f that the ZBPs at the two boundaries do not exhibit the same onset field, which is defined as the field where the zero-bias conductance reaches half of its maximum value. In Fig. 5d, this corresponds to 0.85T on the left and 0.78T on the right side (grey arrows). This observation could be explained by the presence of long-range inhomogeneities that result in a difference in $\Delta\mu$ at the two nanowire ends of ~70 μeV, considering $g = 10$. A possible origin of this inhomogeneity might be a variation in the deformation potential along the length of the hybrid due to a slight bend in the nanowire[56]. At larger values of $\mu$, potential variations are expected to be suppressed due to screening. This might be supported by another data set measured at a larger chemical potential ($V_{SG} \sim 0.5$ V) presented in Supplementary Fig. 28, where we observe the same ZBP onset field at both N–S boundaries. The concomitant evolution as a function of $V_{SG}$ at both nanowire ends is shown in

Supplementary Figs. 28 and 29. This observation might corroborate the signatures of MBS[19,55], but it cannot be regarded as conclusive evidence for truly separated MBS[43].

Figure 5b,c show differential conductance line-cuts, which reveal a zero-bias conductance close to $2e^2/h$ for the ZBP at the left boundary of the device, as highlighted in Fig. 5d. While ZBP conductance close to $G_0$ has been observed for several N–S junctions, it depends on the fine-tuning of the tunnel barriers, which can be strongly affected by transmission resonances. Experimentally, ZBPs are in general substantially lower than the expected value of $G_0$[42,50]. Theoretical studies recently pointed out that partially or fully overlapping MBS can cause quantised ZBPs, indistinguishable from those resulting from isolated MBS[55,57,58]. Hence, the quantised ZBP conductance is a critical but not sufficient hallmark of MBS[54,58].

## Discussion

The 3-terminal hybrid nanowire devices provide a fundamental tool to study the evolution of the induced superconducting gap in the bulk of the hybrid, where electron- and hole-type bands become inverted at the topological phase transition. There, the closing and reopening of the induced gap are accompanied by the emergence of delocalised MBS, hallmarked by ZBPs at both boundaries of the hybrid nanowire[20]. Here, we demonstrate hard-gap N–S junctions in a magnetic field where only discrete subgap states move to zero energy to form ZBPs at both boundaries and that respond similarly to variations in the chemical potential.

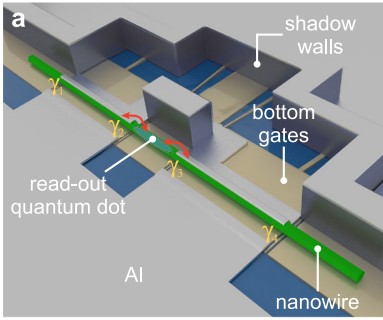

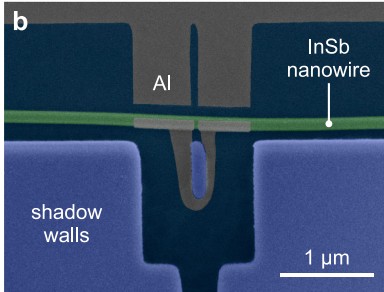

**Fig. 6 Illustration of the proposed Majorana loop qubit. a** Schematic of a single-nanowire loop-qubit device. The presumable locations of the MBS at the boundaries of the two hybrid segments are denoted by $\gamma_i$, where $i \in \{1, 2, 3, 4\}$. The electron parity is fixed due to the finite charging energy of the loop qubit. This configuration offers the desired ground-state degeneracy for a single qubit and can provide information on decoherence and quasiparticle poisoning. **b** False-colour SEM image of an InSb nanowire following the shadow-wall deposition. Two segments of the nanowire are covered with a superconducting 3-facet Al shell. These hybrid segments are interconnected via an Al loop running across the substrate.

While these are critical signatures of MBS, upcoming studies will attempt to correlate the local tunnelling conductance with the evolution of the induced bulk gap via the nonlocal conductance between the two N terminals[18].

Our approach promotes the development of intriguing nanowire-based quantum devices. The ballistic hard-gap N–S junctions together with the thin Al connections across the substrate represent a vital starting point for realising a topological qubit. A qubit implementation with a single read-out loop[7] would allow for measuring the projection of the qubit state on one axis of the Bloch sphere. A schematic of the loop qubit is presented in Fig. 6a. It is made from a single nanowire with two superconductor–semiconductor segments connected via a superconducting loop that encircles a central shadow-wall pillar. Bottom gates at the centre of the device are used to define a read-out quantum dot in the nanowire with tunable tunnel couplings to the MBS denoted as $\gamma_2$ and $\gamma_3$ in the schematic. Parity read-out will be performed by measuring the quantum capacitance via radio-frequency gate reflectometry[6,7,59]. In Fig. 6b, we present an exemplary realisation of the basic elements of such a device via the shadow-wall technique. It comprises a superconducting loop to provide a connection for the exchange of Cooper pairs that acts as a blocker for quasiparticle transport between the two hybridised nanowire segments. The shadow-wall technique is ideally suited to realise these superconducting interconnects across the substrate for multi-terminal devices without the need for post-interface fabrication.

## Methods

**Nanowire growth**. The InSb nanowires are grown on InSb (111)B substrates covered with a pre-patterned SiN$_x$ mask via metalorganic vapour-phase epitaxy

(MOVPE). These nanowires are not grown on top of InP stems but nucleate instead directly on the growth substrate at Au catalyst droplets[30]. The investigated nanowires have an average diameter of 100 nm, which is controlled by the Au droplet size and the growth mask openings, and a typical length in the order of 10 μm.

**Device fabrication**. Bottom gates are fabricated on Si/SiO$_2$ substrates via dry-etching of W thin films, which are subsequently covered by Al$_2$O$_3$ gate dielectric via atomic layer deposition (ALD). Shadow walls of ~600 nm height are created via reactive-ion etching of thick layers of Si$_3$N$_4$ formed via plasma-enhanced chemical vapour deposition (PECVD). Using a micromanipulator, individual nanowires are placed deterministically next to the shadow walls. The native oxide of the nanowires is removed via atomic hydrogen radical cleaning (see Supplementary Note 1). The Al thin films are deposited by evaporation under a shallow angle that forms continuous contacts between the nanowires and the substrate and creates segments on the chip that are electrically isolated from one another. This allows to immediately cool down the devices without the need for additional post-interface fabrication steps. We have not observed a decreased stability or performance of devices that were made with an extra fabrication step to create N contacts. We attribute this to the fact that the hybrid segments are not directly exposed and resist baking is avoided during the fabrication of the contacts.

**TEM analysis**. The cross-sectional lamellae for TEM are prepared using the focused ion beam technique using a Helios G4 UX FIB/SEM from Thermo Fisher Scientific after capping the devices with a protective layer of sputtered SiN$_x$. TEM analysis is carried out at an acceleration voltage of 200 kV with a Talos electron microscope from Thermo Fisher Scientific equipped with a Super-X EDX detector.

**Transport measurements**. Electrical transport measurements are carried out in dilution refrigerators equipped with 3-axes vector magnets. The base temperature is ~15 mK, corresponding to an electron temperature of ~30 mK measured with a metallic N–S tunnel junction thermometer. The sample space is evacuated by a turbomolecular pump for at least 1 day prior to the cool-down to remove surface adsorbates that limit the device performance. Conductance measurements are performed using a standard low-frequency lock-in technique. For voltage-bias measurements, the excitation voltage is $V_{AC} \leq 20$ μV at a lock-in frequency of at least 20 Hz. For all two-terminal conductance measurements we only subtract setup-related series resistances without making any assumptions about additional contact resistances of the metal–semiconductor interface. Current-driven measurements are carried out in a four-point configuration.

After taking the data, we became aware of the relatively low bandwidth of the employed current-to-voltage amplifiers. Hence, we recalibrated the lock-in data via a mapping according to the measured DC conductance that does not suffer from any bandwidth limitations and is insensitive to the reactive response of the circuit (Supplementary Note 4).

**Superconducting gap extraction**. The BCS–Dynes term is given by a smeared BCS density of states with the broadening parameter $\Gamma$[60]:

$$\frac{dI_{SD}}{dV_{SD}}(V_{SD}) = G_N \text{Re}\left[\frac{eV_{SD} - i\Gamma}{\sqrt{(eV_{SD} - i\Gamma)^2 - \Delta_{ind}^2}}\right].$$

For all of our N–S devices, the fit of the BCS–Dynes term yields typical broadening parameters of <10 μeV. The model by Blonder, Tinkham and Klapwijk (BTK) incorporates the transition between BCS tunnelling and Andreev reflection in the open channel regime[21]. Fits of the BCS–Dynes term and of the BTK model to the N–S junction data (including the data in Fig. 4b) are presented in Supplementary Note 4.

The subgap conductance for a ballistic N–S junction with a single subband, where the transport is dominated by Andreev processes, has been described by Beenakker[22]. At a large enough chemical potential[61], it is given by

$$G_S = \frac{4e^2}{h}\frac{T^2}{(2-T)^2} = 2\frac{G_N^2}{(2G_0 - G_N)^2},$$

where the transmission probability, $T$, has been substituted with the normal-state conductance, $G_N$, in units of $2e^2/h$. This function is plotted together with the measured data in Fig. 4e.

## Data availability

The data that support the plots within this paper and other findings of this study are available at https://doi.org/10.5281/zenodo.5034524.

## Code availability

The code used to calculate Andreev transport in Josephson junctions is available from the corresponding author upon reasonable request.

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

## Acknowledgements

We are grateful to Olaf Benningshof for valuable technical support and to Emrah Yücelen, Gijs de Lange, Bernard van Heck, Andrey E. Antipov and Jay D. Sau for fruitful discussions. We thank Morteza Aghaee for support with dielectric deposition and TNO for providing access to their cleanroom facilities. This work has been financially supported by the Dutch Organization for Scientific Research (NWO), the Foundation for

Fundamental Research on Matter (FOM) and Microsoft Corporation Station Q. M.P.N. acknowledges support within the POIR.04.04.00-00-3FD8/17 project as part of the HOMING programme of the Foundation for Polish Science co-financed by the European Union under the European Regional Development Fund.

## Author contributions

S.H., M.Q.P., F.B., N.d.J., P.A., K.v.H. and L.P.K. conceived the experiment. S.H., M.Q.P., F.B., N.v.L., G.P.M., J.S. and M.A.Y.v.d.P. contributed to the fabrication and/or electrical transport measurements of the devices. S.H. and F.B. analysed the transport data. A.F. fabricated the substrates. M.P.N. performed numerical simulations of the MAR processes. M.A. made critical upgrades to the equipment and provided technical support. G. B., S.G. and E.P.A.M.B. carried out the nanowire synthesis. K.L. prepared the FIB lamellae. S.K. performed the TEM analysis. S.H., M.Q.P. and F.B. wrote the manuscript. All authors provided critical feedback. L.P.K. supervised the project.

## Competing interests

The authors declare no competing interests.
