## [Peer Review File · Nature Communications]

REVIEWER COMMENTS

Reviewer #1 (Remarks to the Author):

The manuscript by Heedt et al, reports a novel fabrication scheme for realizing semiconductor/superconductor hybrid nanowire devices. The method relies on clever preparation of the device substrate, manipulation of nanowires, and a new hydrogen cleaning procedure for achieving clean interfaces between the nanowires and superconductors. The method minimizes or eliminates the process steps required after establishing the semiconductor/superconductor interface which is known to be crucial for device performance. Compared to recent developments of shadow-approaches which eliminate certain process steps by combining shadow structures directly with the growth of the nanowires, the present method enables a considerable flexibility in device design and also opens up for unique and important device geometries as electrical connection to the superconductor shells can be achieved. The merits of the method are backed up by comprehensive low temperature electrical characterization of devices and a substantial and detailed supplementary information providing all necessary details. The invention of this new approach and the presented results constitute a significant step forward for the field and I highly support the publication of this work in Nature Comm. The manuscript could be further improved addressing the following points.

- 1) As stated in the abstract the new shadow-wall procedure "offers substantial advances in device quality and reproducibility". Reproducibility is, however, is not systematically considered in the manuscript. Could the authors include a discussion of the reproducibility of the results of shadow-wall devices compared to conventional devices. Also, it is not clear what the "substantial advance in device quality" refers to, and I suggest the authors include this discussion as well maybe as a part of the summary.
- 2) Considering the characterization of the shadow-wall Josephson Junction in Fig. 3 the authors state that the critical current of 90 nA is "remarkable". On p1 the I_c value is quoted as "exceptionally high" compared to previous studies. Could the authors elaborate on in which way this value is considered remarkable and how it compares to the literature on nanowire JJ taking into account the influence of the normal-state resistance. Is the result a consequence of the shadow-wall fabrication scheme?
- 3) Also considering the JJ, the authors attribute the observed I_{sw} / I_r hysteresis to the dynamics of an underdamped junction in the RCSJ model. If possible, I suggest the authors provide estimates of the parameters of the effective RCSJ model. Can heating effects be excluded as a cause of the hysteresis?
- 4) Considering the N-S characterization (Fig. 4) and the "two-ended N-S-N" devices in Fig. 5, the fabrication process includes a step of conventional lithography / milling. The processing seen by the nanowire thus seem closer to that of conventional devices based on epitaxial growth of hybrid nanowires or on shadow patterning of superconductors during nanowire growth. This should be clearly stated. The manuscript states that this post-interface fabrication step is "optional" – if there is a way to include the normal contacts in the scheme of shadow-defined fabrication it would be relevant to include it here. Could the authors comment on any differences in terms of stability / reproducibility between these devices and the devices which do not see any post processing? Also, the characterization of the induced superconductivity in Fig. 4 is very similar to the original report of hard-gap in epitaxial nanowires (Chang et al., Nature Nanotechnology 10, 232 (2015)) and this should be cited.
- 5) In connection to Fig. 5 the author state that "we see in Figs. 5e,f that the ZBPs at the two boundaries do not exhibit the same onset field". Could the authors define the onset-field and provide a

measure of the observed difference. Also it would be relevant to discuss what is considered "long-range inhomogeneity" (long compared to what?) and, if possible, relate this quantitatively to the difference in onset field? Finally, could the authors comment on the possible origin of such long range inhomogeneity given the new delicate fabrication scheme?

/Thomas Sand Jespersen
Niels Bohr Institute
Denmark

Reviewer #2 (Remarks to the Author):

The manuscript reports on a shallow-wall lithography technique for fabrication of semiconductor nanowire-superconductor hybrid devices with high-quality interfaces promising for creating Majorana bound states and Majorana qubits. The technique has been described in detail and successfully demonstrated with transport measurements of a few devices made with the technique. Thus, the manuscript is of interest to the topological quantum physics and devices community and the nanotechnology community, and could be published in Nature Communications. However, I would not recommend publication of the manuscript in its present form. A few concerns which need to be addressed for improvement are given below.

- In abstract, the first sentence is too strong or it addresses a too narrow field. There must be several other schemes to proceed with realization of a topological qubit. Please consider to rephrase the sentence.
- Figure 1 is an important figure for illustrating the Shall-wall technique, which should make all easy to understand. However, the blow-up circle of Figure 1c does not precisely describe the green circle part in Figure 1c.
- In the description for device structures, it would make much easier to the reader that the thicknesses of different layers, such as bottom gates, Al₂O₃, etc., as well as the cross-section sizes of nanowires are given in the text or in the relevant figure captions.
- For devices with normal contacts included, some more description about how these contacts are made, what are layer thicknesses, etc., should be given.
- It is not clear that the devices measured for this work are all on the same chip or on different chips. The authors should make the description about the differences (if there are) between these measured devices clear.
- In the measurements presented for JJs, some interesting features are seen in Figure 3. Some these features, like above gap faint lines are explained in the text. But, there are string features for which no clear explanation are presented. For example, what are the origin of the high resistance lines seen at around $I_{sd} \sim |0.2| \mu A$ in Fig. 3a and what is the physics behind the temperature dependence of these features? Why is the T_c goes much higher than that for Al?
- In Fig. 3b, are observed low bias peaks the higher order MAR peaks? Can these low bias peaks also be fitted satisfactorily by theory together with high bias peaks?
- For Fig. 3c, could the authors commend on the splitting features at around V_{sd} around 0.5 mV and

0.25 mV? What are the physical origins of these splittings and their complicated magnetic field dependences? What happens at around 0.4 T in parallel magnetic field, where it is observable that lines come to together as well as move apart in complicated ways? Similar complicated features are also shown in Figs. S8 and S9.

- For devices shown in Figs. 4a and 5a, again, it is good to add description about how are Ti/Au contacts are made and what are their thickness. What are the impact of this fabrication procedure on the device part made by shallow-wall technique? Note a misprint in caption of Fig. 4c, i.e., (a) should be (b). Note also a misprint in Fig. 4e, i.e., 650 μeV should read 650 μV .

- in Section D, it states that at low V_{tg} , subgap conductance is suppressed by about two orders of magnitude when compared to normal-state conductance. Is this enough to claim achieving a hard gap in the hybrid nanowires?

- In Section E, to my understanding, no convincing evidence on the correlation between ZBPs at the two ends has been provided. Thus, I would not say that the title of the section is appropriate here. Could you consider a revision?

- In Fig. 6b, I would not think that this device will be a successful three-terminal device, since the SC loop in the device is hard to be bounded (or am I wrong?). The authors should commend on this or replaced the device picture with a more proper one.

- Some typeset issues: A more important one is that please always use a "-" in the phrase like "normal metal-superconductor", or always use a "/" in such phrases.

Reviewer #3 (Remarks to the Author):

The manuscript presents a novel, unconventional technique to fabricate hybrid semiconductor-superconductor nanowire devices without the need for resist coating and etching processes that could degrade the device surface. The technique, based on aluminum angle evaporation through predefined shadow walls, is original and well characterized by means of multiple inspection techniques based on SEM and TEM. An abundance of illustrative images is provided both in the main text and in the supplementary material.

In order to validate their fabrication technique, the authors have performed low-temperature transport experiments on a set of two-terminal and three-terminal hybrid devices. Fig. 3, as well as Figs. S5 & S6, show data for Al-InSb nanowire-Al devices with a relatively short 'channel' length of 110-150 nm, comparable to the nanowire diameter.

Superconductor-semiconductor junctions can show relatively high transparency, validating the surface cleaning process by atomic hydrogen radical cleaning. From the fitting of MAR features, the authors extract transparencies close to unity for the first subband. In Fig. S5, conductance quantization is claimed at large finite bias, i.e. well above the superconducting gap ($V_{\text{SD}} = 10 \text{ mV} \gg \Delta$). This claim is questionable though: the steps are barely visible (which is quite typical for nanowire devices); to my understanding, the series resistance cannot be independently measured.

Fig. 4 shows data for a normal metal-InSb nanowire-Al device. Suppressed sub-gap conductance is shown in the low-transparency regime. Upon increasing the voltage on tunnel gate (hence lowering the tunnel barrier), a step-like increase of the sub-gap conductance is observed. The above-gap

conductance exhibits a quasi-plateau around the conductance quantum. Simultaneously, the zero-bias conductance shows a peak at about twice the conductance quantum, followed by a pronounced dip on the right side of Fig. 4d. Fig. 4 is used to claim ballistic transport in the presence of Andreev reflection, but I have several concerns about that: 1) Based on the discussion in Supplementary Section IV, data in Fig. 4 are plotted following the subtraction of a series resistance. Surprisingly, I could not find that mentioned anywhere in the main text, and the value for the subtracted series resistance was not specified. To claim $G=2e^2/h$, this point is crucial. 2) To access the normal-type regime, the conductance is measured at relatively high bias voltage, where the effects of disorder-induced scattering and possible Coulomb-blockade effects are largely washed out (btw, the same is true for the data of Figs. S5 & S6). Not surprisingly, the low-bias conductance shows much stronger modulations. 3) a strong conductance dip is observed for $V_{TG} \sim 0.8$ V. The explanation in terms of "inter-subband scattering" is not convincing. To begin with, the authors refer to the onset of conduction through the second subband, but this cannot be seen from the data shown. Fig. 5 shows data for a three-terminal device (one superconducting contact covering a 1- μ m-long segment of the nanowire, and two normal-type contacts at the edges). Around 1 T, zero-bias conductance peaks develop at both edges with no apparent correlation. A second data set is shown in Fig. S24. The peak heights observed at both edges are relatively high, possibly close to $2e^2/h$ (once again, I wasn't able to find any value of the subtracted series resistance. The authors only refer to an uncertainty of 0.5 k Ω on the series resistance, but how can they say that?).

Overall, based on the results of the different transport experiments, I do not see a clear benefit from the presented fabrication technique over previously used approaches. Owing to the avoided device processing, I would have expected some clear improvement in the transport properties of the nanowires, e.g. a clearer evidence of ballistic transport, but what is shown does not go beyond the state-of-the-art. The same group and other groups have reported results of comparable level, e.g. Zhang et al., *Nature* 556, 74–79(2018); Gül et al, *Nature Nanotechnology*, 13, 192–197(2018); Abay et al. *Nano Letters*, 13, 3614 (2013); Xiang et al. *Nature Nanotechnology* volume 1, 208–213(2006). For this reason, I would not recommend publication in *Nature Communications*.

Response to the reviewers' comments:

We very much appreciate the thorough analysis of our manuscript and we thank you for the commendatory words and the valuable feedback that you have provided. All questions and comments are clear and will be addressed in the following point by point. Below, we show the referees' remarks in black and our response in blue. The amendments in the revised manuscript and in the Supporting Information are highlighted in orange.

Reviewer #1 (Remarks to the Author):

The manuscript by Heedt et al, reports a novel fabrication scheme for realizing semiconductor/superconductor hybrid nanowire devices. The method relies on clever preparation of the device substrate, manipulation of nanowires, and a new hydrogen cleaning procedure for achieving clean interfaces between the nanowires and superconductors. The method minimizes or eliminates the process steps required after establishing the semiconductor/superconductor interface which is known to be crucial for device performance. Compared to recent developments of shadow-approaches which eliminate certain process steps by combining shadow structures directly with the growth of the nanowires, the present method enables a considerable flexibility in device design and also opens up for unique and important device geometries as electrical connection to the superconductor shells can be achieved. The merits of the method are backed up by comprehensive low temperature electrical characterization of devices and a substantial and detailed supplementary information providing all necessary details. The invention of this new approach and the presented results constitute a significant step forward for the field and I highly support the publication of this work in Nature Comm.

Thank you for the comprehensive review and the appreciative remarks. The critical feedback was very valuable and has improved the manuscript. In our resubmission, we have addressed all your comments and suggestions.

The manuscript could be further improved addressing the following points.

1) As stated in the abstract the new shadow-wall procedure “offers substantial advances in device quality and reproducibility”. Reproducibility is, however, is not systematically considered in the manuscript. Could the authors include a discussion of the reproducibility of the results of shadow-wall devices compared to conventional devices. Also, it is not clear what the “substantial advance in device quality” refers to, and I suggest the authors include this discussion as well maybe as a part of the summary.

We appreciate this comment very much. If we want to compare shadow-wall lithography with conventional methods, it is necessary to distinguish between (a) conventional lift-off junctions, (b) conventional etched junctions and (c) conventional shadow junctions. Common to all these approaches is that devices rely on singular, customized designs and require SEM imaging of each nanowire for accurate alignment via electron-beam lithography. In contrast, our method allows for convenient blind fabrication (i.e., without prior imaging and alignment) of many nominally identical devices. This is crucial for conveniently reproducing very similarly looking and performing samples.

Conventional method (a) produces lithographically defined hybrid devices with similar reproducibility of the geometric dimensions as our technique (e.g., overlay accuracy between gates and junctions). Method (a) is incompatible with atomic H cleaning (since it is not resist-free) and requires ex-situ NH_4S_x wet-chemical treatment as well as in-situ Ar plasma cleaning of the InSb surface to remove the native oxide. Here, the main

limitation is the interface quality resulting from this cleaning procedure. Gül et al. (Nano Lett. 17, 2017), Nilsson et al. (Nano Lett. 12, 2012), and Li et al. (Sci. Rep. 6, 2016) followed that approach and had to introduce a thin layer of Ti at the interface to facilitate a high interface transparency, which is detrimental for the gap hardness. Conventional InSb/Al Josephson junction devices by Gül et al. show an induced gap of only 150 μeV due to the much larger Al thickness, which survived up to 25 mT, whereas in our devices the critical field is roughly 50 times larger. Those Josephson junctions exhibited I_{sw} of up to 10 nA, which yields an $I_{sw}R_N$ product 7 times smaller than in our devices. Hybrid nanowire devices made via method (a) typically do not demonstrate hard gaps and N-S junctions show quite substantial subgap conductance (cf. also Zhang et al., Nat. Commun. 8, 2017 or Yu et al., Nat. Phys., 2021).

Method (b) relies on high-quality epitaxial InSb/Al nanowires, but the junctions are formed by selectively removing the Al from the semiconductor. However, selective etch conditions for Al on InSb have not yet been found (as discussed by Khan et al., ACS Nano 14, 2020). The reproducibility of etched junctions is typically very poor, and the metal etching is often uncontrollable, causing substantial disorder at the junction area (see also M. W. A. de Moor, Ph.D. thesis, Delft University of Technology, 2019), which can deteriorate signatures of ballistic transport. To highlight this point, we have added a sentence in the “Shadow-Wall Lithography” section:

“Common to those methods is that the hybrid nanowires are removed from the growth substrate following the evaporation and undergo several post-interface fabrication steps such as alignment via scanning electron microscopy (SEM), electron-beam lithography involving resist coating, or etching. **The latter in particular degrades the electrical device performance compared with shadowed junctions [Khan2020]. Moreover,** hybrid devices are prone to degradation. High-temperature processing ...”

Method (c) represents the state-of-the-art for fabricating hard-gap, low-disorder InSb/Al hybrid devices, but it does not offer the same inherently good alignment between the edges of the superconductor and the gates as our technique. A hexagonal nanowire covered with Al on 3 facets can have 6 rotational configurations on the substrate (4 of which are unique; cf. illustration below that is also included as the new Fig. S3 in the Supplementary Information), which are all possible orientations for method (c). This affects the gate geometry, the contacting via normal-metal leads, and the overall electrostatics of the hybrid nanowires. However, our shadow-wall method always enforces the same orientation (panel a below), efficiently removing this random variation between devices. Conventional shadow junctions also cannot be combined with bottom gates (shadow junctions and bottom gates are extremely hard to align) and desirable high-temperature processing of the top-gate dielectric (e.g., atomic layer deposition) cannot be performed due to the low thermal budget.

Unique orientations of the Al-covered facets on hexagonal nanowires.

Shadow-wall lithography offers the same interface quality as state-of-the-art in-situ deposition methods, which allow for hard gaps, but it effectively removes the variations inherent to other methods and introduces considerable flexibility and fundamentally new device designs. Our approach reduces the risk of ageing of the hybrid interface by requiring at most a single fabrication step and it provides a large number of nominally identical devices. By design, all nanowires are aligned along the same direction, which is a bonus feature of the shadow-wall technique, as it simplifies the alignment of the magnetic field along the wire axis. This is a key requirement to search for a topological phase in all available hybrid nanowire devices on a given chip.

“Device reproducibility” also refers to the reproducibility of the device geometry as well as the yield of samples of similar quality. To illustrate the low variability in the device dimensions, we have added the new Fig. S4 to the Supplementary Information which shows box plots of the extracted Josephson junction lengths for in total 34 devices fabricated on 3 different substrates (the SEM images of all samples on chip U51 have been added in the new Fig. S6). These box plots of the junction lengths demonstrate very narrow distributions and show that the median values of two nominally identical substrates (U51 and U55) differ by only 4 nm. This plot emphasizes that many nearly identical devices can be repeatably produced in a single deposition step.

We demonstrate the reproducibility of the shadow-wall technique by adding the new Fig. S16 to the Supplementary Information. There, we present conductance–bias-voltage traces in the tunnelling regime for 7 Josephson junction devices with similar diameters (~ 100 nm) from 3 different substrates that all show a very similar magnitude of the induced gap with a standard deviation of only $10 \mu\text{eV}$. The same is true for all the N-S junctions studied in this work (cf. Figs. 4 & 5 and Figs. S23 & S24), which consistently demonstrates a hard gap of $230\text{-}240 \mu\text{eV}$ and highlights a reproducible interface transparency. The N-S junctions also demonstrate a complete absence of subgap states at zero field, which is further testament to the high device quality and homogeneity. We have added the following sentence in Supplementary Section IV.C:

“The two N-S junctions presented in Fig. 5 of the main text (devices 4 and 5) also exhibit comparable values of the induced gap of $\Delta_{\text{ind}} \sim 230\text{-}240 \mu\text{eV}$.”

As proposed by the referee, we have included a discussion of the device quality and reproducibility. In the introduction we first define more clearly what we mean by “high-quality devices”:

“Here, we introduce a novel fabrication technique that resolves these challenges and provides high-quality hybrid quantum devices, **reflected by the absence of chemical intermixing, a high interface transparency and hard induced gaps, while** involving minimal nanofabrication steps compared with previously established methods [10,11].”

In the Supplementary Information (Supplementary Section I.F), we have added a comprehensive discussion of the device reproducibility and the qualitatively new aspects of our technique, which is referenced in the Introduction:

“Similar advances in quality and reproducibility (**Supplementary Section I.F**) were made possible by the reverse fabrication process established for carbon nanotube devices [Cao2005].”

2) Considering the characterization of the shadow-wall Josephson Junction in Fig. 3 the authors state that the critical current of 90 nA is “remarkable”. On p1 the I_c value is quoted as “exceptionally high” compared to previous studies. Could the authors elaborate on in which way this value is considered remarkable and how it compares to the literature on nanowire JJ taking into account the influence of the normal-state resistance. Is the result a consequence of the shadow-wall fabrication scheme?

In the following table, we compare our Josephson junctions' transport properties with results from similar devices on InSb nanowires. To account for the influence of the normal-state resistance, we also display the $I_{sw}R_N$ product. Moreover, since Gül et al. also considered NbTiN as a superconductor, we also included this product normalized by the gap size. Our findings stand out compared to the previous studies with regard to both of these quantities and with regard to the magnitude of the switching current itself.

In conclusion, we are convinced that the superior results are related to the cleanliness of the InSb/Al interface obtained by the in-situ hydrogen cleaning prior to the shadow deposition, but also to the absence of fabrication steps after the realization of the InSb/Al interface.

Reference	Materials	Length (nm)	Δ_{ind} (μeV)	T (mK)	Max. I_{sw} (nA)	$I_{sw}R_N$ (μV)	$eI_{sw}R_N/\Delta_{ind}$
this work	InSb/Al	115	235	30	90	110	0.47
Gül et al.	InSb/Ti/Al	150	150	250	10	15	0.10
	InSb/NbTiN	150	750	50	40	160	0.21
Nilsson et al.	InSb/Ti/Al	30	150	30	5	34	0.23
Li et al.	InSb/Ti/Al	60	150	10	12	10-30	0.20

Comparison with previous reports on InSb nanowire-based Josephson junctions.

To clarify this comparison, we completed the sentence in the Introduction:

“These junctions exhibit gate-tunable supercurrents of up to 90 nA, which is exceptionally large for InSb/Al nanowires **compared to previous works on InSb Josephson junction devices** [9, 16, 17].”

Moreover, we added a sentence to better quantify how our results compare to the literature in the section “Highly Transparent Josephson Junctions” after the sentence ending with “... current fluctuations.”:

“We note that the magnitude of I_{sw} as well as the normalized quantity $eI_{sw}R_N/\Delta_{ind}$ are significantly larger than in previous reports on InSb Josephson junctions [Gül2017, Nilsson2012, Li2016].”

3) Also considering the JJ, the authors attribute the observed I_{sw} / I_r hysteresis to the dynamics of an underdamped junction in the RCSJ model. If possible, I suggest the authors provide estimates of the parameters of the effective RCSJ model. Can heating effects be excluded as a cause of the hysteresis?

We thank the reviewer for this comment. We have reconsidered the cause of the hysteresis of the $V_{SD}-I_{SD}$ traces, and we now attribute it indeed to Joule heating at the junction in the voltage-state. As initially described by Courtois et al., Phys. Rev. Lett. 101, 2008, this is much more likely to be the cause in SNS devices with such a low capacitance (less than 10 aF for our junctions). The temperature dependence in Fig. 3a supports this hypothesis. There, we notice that I_r stays rather constant up to 600 mK, while I_{sw} decreases till matching the retrapping current I_r . The disappearance of the hysteresis for $T > 600$ mK is consistent with an enhanced thermalization mediated by phonon coupling. Considering a realistic capacitance in the 10 aF range, a switching current of 88 nA, and the normal-state resistance of 1.25 k Ω , we obtain the Stewart-McCumber parameter of $\beta = 2eI_{sw}R_N^2C/\hbar \sim 0.004 \ll 1$, from which we would expect a non-hysteretic $V_{SD}-I_{SD}$ characteristic in the absence of Joule heating. Hence, the capacitance of these nanowire junctions is too small to explain the hysteresis to originate from a Josephson junction in the underdamped regime.

We have changed the sentence in the section “Highly Transparent Josephson Junctions” regarding the origin of the hysteresis, also including an additional reference to the literature (Courtois et al. Phys. Rev. Lett. 101, 2008):

Original version:

“At low temperatures ($T < 0.6$ K), the hysteretic behaviour of the asymmetric $V_{SD}-I_{SD}$ traces indicates that the junction is in the underdamped regime according to the model of resistively and capacitively shunted junctions. Above 0.6 K, the thermal activation washes out the asymmetry of the traces.”

New version:

“At low temperatures ($T < 0.6$ K), the hysteretic behaviour of the asymmetric $V_{SD}-I_{SD}$ traces **is caused by self-heating of the junction. This effect disappears at higher temperatures ($T > 0.6$ K), which can be attributed to enhanced thermalization via electron-phonon coupling [Courtois2008].**”

4) a) Considering the N-S characterization (Fig. 4) and the “two-ended N-S-N” devices in Fig. 5, the fabrication process includes a step of conventional lithography / milling. The processing seen by the nanowire thus seem closer to that of conventional devices based on epitaxial growth of hybrid nanowires or on shadow patterning of superconductors during nanowire growth. This should be clearly stated. The manuscript states that this post-interface fabrication step is “optional” – if there is a way to include the normal contacts in the scheme of shadow-defined fabrication it would be relevant to include it here.

Following the referee’s advice, we have clarified that due to this single fabrication step the processing of the N-S junctions resembles the N contact formation in conventional shadow junctions (see new version below: “... similar to the contacting of conventional shadow junctions ...”). However, it is important to point out that this step is only minimally invasive. We don’t perform selective etching steps or high-temperature processing (resist degassing is performed by vacuum pumping instead of conventional baking and the dielectrics are already deposited) and the single lithography step used to form the N contacts does not entail electron-beam expose of the sensitive hybrid nanowire segment. In Supplementary Section I.C, we have included additional information on the N contact formation:

“For devices with additional Ti/Au normal-metal contacts, such as the ones presented in Figs. 4 and 5 of the main text, an extra post-interface fabrication step is included. It consists of EBL patterning **(solvents are removed from the resist via vacuum pumping instead of conventional resist baking to accommodate the low thermal budget), ...**”.

Even when this extra step is included, ageing/chemical intermixing at the semi-/superconductor interface (see also S. Gazibegovic, Ph.D. thesis, Eindhoven University of Technology, 2019) can be largely avoided since the N contact deposition takes significantly less time and requires less thermal budget than the complete device fabrication (including N contacts, gate dielectric and top-gate metal).

The term “optional” was explained in the Methods section, but we have now incorporated this explanation in the main text. We have added a second option based on our shadow-wall technique that requires metal deposition from two different angles, which we describe in detail in a separate manuscript (Borsoi et al., arXiv:2009.06219, 2020). In Section II.D, we now present two alternative solutions offered by the shadow-wall technique to entirely omit any post-interface fabrication step:

Original version:

“An exemplary N-S device is depicted in Fig. 4a. Here, the N contact to the InSb nanowire was formed in an optional post-interface fabrication step.”

New version:

“An exemplary N-S device is depicted in Fig. 4a. Here, the N contact to the InSb nanowire was formed in a **post-interface fabrication step, similar to the contacting of conventional shadow junctions (Supplementary Section I.C). Alternatively, Al leads that are defined by the shadow walls – microns away from the N-S junction – can serve as N contacts but require additional bottom gates to render all nanowire segments fully conducting (cf. Fig. 1b). Another option to fabricate N contacts in situ involves using two deposition angles, which we describe in detail elsewhere [Borsoi2020].**”

4) b) Could the authors comment on any differences in terms of stability / reproducibility between these devices and the devices which do not see any post processing?

The devices made without any post-interface fabrication presented in our manuscript are the Josephson junctions, while the ones involving a single processing step are the N-S devices. Overall, we did not observe variations in transport features, stability, and quality (reflected in variations of the induced gap and signatures of ballistic transport) between these two types of devices. We attribute this to the fact that the fabrication of the N contacts takes up less than two days prior to the cooldown of the samples and that the involved processes are carried out at room temperature, avoiding resist baking. We have included a sentence on this aspect at the end of the Methods section:

“We have not observed decreased stability or performance of devices that were made with an extra fabrication step to create N contacts. We attribute this to the fact that the hybrid segments are not directly exposed and resist baking is avoided during the fabrication of the contacts.”

In the Supplementary Information, we discuss the reproducibility of shadow-wall devices considering the transport properties of 7 Josephson junction devices (new Fig. S16) and 5 N-S junctions. All devices presented here require at most one post-interface fabrication step, which is only minimally invasive since the hybrid InSb/Al segments remain untouched. The fact that the N-S devices do not exhibit any subgap states at zero magnetic field and that the hard induced gaps are well described by the BTK theory, with a gap size similar to the Josephson junctions that did not require any post-interface fabrication, demonstrates that this room-temperature metal deposition step outside of the hybrid segments did not negatively impact the device performances.

4) c) Also, the characterization of the induced superconductivity in Fig. 4 is very similar to the original report of hard-gap in epitaxial nanowires (Chang et al., Nature Nanotechnology 10, 232 (2015)) and this should be cited.

As suggested, we have included the citation of this work as follows:

“The measure of success is a hard induced gap at a finite magnetic field and quantized Andreev enhancement as a signature of ballistic transport [Chang2015, Kjaergaard2016].”

5) In connection to Fig. 5 the author state that “we see in Figs. 5e,f that the ZBPs at the two boundaries do not exhibit the same onset field”. Could the authors define the onset-field and provide a measure of the observed

difference. Also it would be relevant to discuss what is considered "long-range inhomogeneity" (long compared to what?) and, if possible, relate this quantitatively to the difference in onset field? Finally, could the authors comment on the possible origin of such long range inhomogeneity given the new delicate fabrication scheme?

The difference in the onset field might indicate a variation of the chemical potential on the scale of the hybrid device length. If the left junction enters a topological phase at a different chemical potential than the right side, it is more likely that the nonuniformity in the system is long-range rather than short-range.

The difference in the onset fields ΔB_{ZBP} can be quantified by the difference in the field values where the conductance at zero bias voltage reaches half of its maximum value. We find $\Delta B_{ZBP} \sim 0.07$ T, where the onset field on the left side is $B_L = 0.85$ T and the onset field on the right side is $B_R = 0.78$ T. If the two ZBPs are in fact originating from a single delocalized state, we can approximately quantify the chemical potential difference between both sides:

$$\Delta\mu = \sqrt{(g\mu_B B_L)^2 - \Delta_{ind}^2 [1 - (B_L/B_c)^2]^2} - \sqrt{(g\mu_B B_R)^2 - \Delta_{ind}^2 [1 - (B_R/B_c)^2]^2}$$

Assuming $\Delta_{ind} \approx 230$ μ eV, $g \approx 10$, $B_c \approx 1.25$ T, this leads to $\Delta\mu \approx 70$ μ eV. We speculate that this can be attributed to a strain variation along the hybrid due to the slight bend in the nanowire visible in the SEM image. In particular, the deformation potential for InSb is 70 meV per 1% strain (see Vurgaftman and Meyer, J. Appl. Phys. 89, 2001 and Gielen and Mackenzie, Microelectron. Reliab. 62, 2016). A small difference in the deformation potential caused by the slight bending of the nanowire (different strain state on either side of the device) could explain a variation in the chemical potential between the two nanowire ends by about 70 μ eV, reflected in the different onset fields of the ZBPs. What matters here is the *difference* in the deformation potential. This does not mean that one cannot bring both sides into a topological regime but only that the ZBPs do not necessarily coincide. To clarify this reasoning, we have modified the manuscript:

Original version:

"We see in Figs. 5e,f that the ZBPs at the two boundaries do not exhibit the same onset field, which could be explained by the presence of long-range inhomogeneities that result in different values of E_Z^C at the two nanowire ends."

New version:

"We see in Figs. 5e,f that the ZBPs at the two boundaries do not exhibit the same onset field, which is **defined as the field where G reaches half of its maximum value. In Fig. 5d, this corresponds to 0.85 T on the left and 0.78 T on the right side (grey arrows). This observation could be explained by the presence of long-range inhomogeneities that result in a difference in E_Z^C at the two nanowire ends of about 70 μ eV, considering $g = 10$. A possible origin of this inhomogeneity might be a variation in the deformation potential along the length of the hybrid due to a slight bend in the nanowire [Gielen2016].**"

Reviewer #2 (Remarks to the Author):

The manuscript reports on a shallow-wall lithography technique for fabrication of semiconductor nanowire-superconductor hybrid devices with high-quality interfaces promising for creating Majorana bound states and Majorana qubits. The technique has been described in detail and successfully demonstrated with transport measurements of a few devices made with the technique. Thus, the manuscript is of interest to the topological quantum physics and devices community and the nanotechnology community, and could be published in Nature Communications. However, I would not recommend publication of the manuscript in its present form. A few concerns which need to be addressed for improvement are given below.

We thank the referee for the detailed review of our article and the positive verdict on this work. We have addressed all critical points in detail below.

1) In abstract, the first sentence is too strong or it addresses a too narrow field. There must be several other schemes to proceed with realization of a topological qubit. Please consider to rephrase the sentence.

We have rephrased this sentence considering a wider spectrum of applications:

Original version:

“The realization of a topological qubit calls for advanced techniques to readily and reproducibly engineer induced superconductivity in semiconductor nanowires.”

New version:

“Advanced techniques that allow to readily and reproducibly engineer induced superconductivity in semiconductor nanowires enable a wide range of quantum devices, in particular the realization of a topological qubit.”

2) Figure 1 is an important figure for illustrating the Shall-wall technique, which should make all easy to understand. However, the blow-up circle of Figure 1c does not precisely describe the green circle part in Figure 1c.

The difference between the blow-up and the encircled part of the SEM image in Fig. 1c is the fact that the SEM is taken before the Al deposition while the blow-up illustrates the situation following the shallow-angle Al deposition, highlighting the purpose of the small gap in the shadow wall. However, it does illustrate the same segment of the shadow walls. To clarify this difference, we have added two comments in the caption:

“False-colour SEM image of an exemplary sample **prior to Al deposition**. Shadow walls are designated in blue and bond pads, which are enclosed by the shadow walls, are shaded in dark yellow. Gaps are placed at critical locations along the shadow walls (cf. green circle and the illustration in the blow-up **following Al deposition**). This ensures that bond pads with leads are isolated from each other after the Al deposition.”

3) In the description for device structures, it would make much easier to the reader that the thicknesses of different layers, such as bottom gates, Al₂O₃, etc., as well as the cross-section sizes of nanowires are given in the text or in the relevant figure captions.

For devices with normal contacts included, some more description about how these contacts are made, what are layer thicknesses, etc., should be given.

In Section II.D of the main text, we have added a reference to Supplementary Section I.C where these detailed information are provided:

“An exemplary N-S device is depicted in Fig. 4a. Here, the N contact to the InSb nanowire was formed in a post-interface fabrication step (**Supplementary Section I.C**).”

In addition, we have included the most relevant information in the caption of Figs. 4 and 5. In Fig. 4, we added “... underneath the **100 nm wide** InSb nanowire ...”. For clarity, we replaced the SEM in Fig. 4a with the SEM of the sample that was studied in Figs. 4b-e and changed the sentence “In Fig. 4b, we present voltage-bias spectroscopy at such an N-S junction ...” to “In Fig. 4b, we present voltage-bias spectroscopy **of the N-S junction in Fig. 4a ...**”.

In Fig. 5, we added “... correlation device **based on an 80 nm wide InSb nanowire with a 1 μ m long hybrid segment.**”.

4) It is not clear that the devices measured for this work are all on the same chip or on different chips. The authors should make the description about the differences (if there are) between these measured devices clear.

This is an important point. We have added a note in Section II.E to highlight that the samples in Figs. 4 and 5 are in fact fabricated together on the same chip:

“As shown in Fig. 5a, optional Ti/Au contacts are again added at both nanowire ends **in the same fabrication run and on the same substrate as the sample in Fig. 4.**”

In the Supplementary Information, we have included a new table (Table III) to summarize all N-S devices studied here. As can be seen in Table I, the Josephson junction devices are in fact from different chips.

5) In the measurements presented for JJs, some interesting features are seen in Figure 3. Some these features, like above gap faint lines are explained in the text. But, there are string features for which no clear explanation are presented. For example, what are the origin of the high resistance lines seen at around $I_{sd} \sim |0.2| \mu\text{A}$ in Fig. 3a and what is the physics behind the temperature dependence of these features? Why is the T_c goes much higher than that for AI?

We thank the reviewer for highlighting that the nature of these features has not been clearly explained. At low temperature, for currents larger than the switching current, the voltage developed across the junctions remains still below $2\Delta/e$ (within the range of the measurement). In this regime, transport of quasiparticles is mediated by multiple Andreev reflections, which cause the sequence of peaks and dips in the resistance as a function of current bias. Because these features are explained in the second part of the Section “Highly Transparent Josephson Junctions” and in the Supporting Information, we clarified this point by adding the following sentence in the caption of Fig. 3a:

“The peaks at $I_{SD} > I_{sw}$ arise from quasiparticle transport via multiple Andreev reflections.”

The peak at the transition region is related to the temperature at which the junction becomes non-hysteretic. This is illustrated by the following schematic:

Explanation of the differential resistance peak in hysteretic Josephson junctions.

The differential resistance is measured via a standard lock-in technique, which involves applying an AC excitation voltage I_{AC} . In the vicinity of I_{sw} , the low-frequency excitation causes a switch from the supercurrent branch to the resistive branch, which is reflected in the steplike increase in the differential resistance (bottom left panel). However, only in the non-hysteretic junction (right panels) the switch at I_{sw} is reversible and the differential resistance measured via the lock-in amplifier exhibits a peak right at the switch (bottom right panel). In hysteretic junctions, these peaks can only occur when the differential resistance is extracted by numerically differentiating the measured DC voltage instead of using a lock-in technique.

T_c is much higher than for bulk Al since we are studying a thin film of Al where the gap and the critical temperature are strongly thickness dependent. In addition, the gap we are probing here is the induced gap. As pointed out initially by Meservey and Tedrow (Fig. 3 of J. Appl. Phys. 42, 1971), the critical temperature T_c of thin Al films is much higher than for bulk Al, and it goes as $T_c \propto d^{-1}$, where d is the film thickness. We make this point clear by extending this sentence:

“The blue region ($R = 0 \Omega$) denotes the superconducting phase, which persists up to ~ 1.8 K, **consistent with the enhanced superconducting critical temperature for thin films with respect to bulk Al [Meservey1971].**”

6) In Fig. 3b, are observed low bias peaks the higher order MAR peaks? Can these low bias peaks also be fitted satisfactorily by theory together with high bias peaks?

The low-bias peaks are indeed attributed to higher order MAR processes. We have extended the fits up to the 5th order in MARs that extend also to lower bias voltages (previously $|V_{SD}| > 100 \mu V$, now $|V_{SD}| > 75 \mu V$). We have updated Fig. 3b and Fig. S8 with these new fits. The extracted values for T_i and Δ_{ind} change only slightly. Even higher orders are less well described by the theory due to the loss of phase coherence, which is not

considered by the model. The model will reproduce the experiment when all its assumptions are realized in the sample: i.e., the transport is coherent, there is no mode mixing due to Andreev reflection, the gap is hard, and the junction is in the short-junction limit. Breaking any of them will result in deviation of the measured data from the theoretical prediction, especially in the case where the bias is low, and each transport process requires many sequential quasiparticle transport processes. Low-bias transport in realistic samples exceeds the limits of the model that we apply (cf. Supplementary Section III).

7) For Fig. 3c, could the authors comment on the splitting features at around V_{SD} around 0.5 mV and 0.25 mV? What are the physical origins of these splittings and their complicated magnetic field dependences? What happens at around 0.4 T in parallel magnetic field, where it is observable that lines come to together as well as move apart in complicated ways? Similar complicated features are also shown in Figs. S8 and S9.

We appreciate this remark. In the revised version of the manuscript, we have added a section to the Supplementary Information where we discuss the field evolution in detail considering the presence of a subgap state close to the gap edge. To shed light on the rich magnetic field dependences presented in Figs. 3c, S11, and S14, we have extended the existing discussion in Supplementary Section III.B. In the original submission, we introduced a model describing the conductance of the device in the presence of multiple Andreev reflections and subgap states. This enabled us to describe the experimental traces at 0 and 0.2 T presented in Fig. S18. Triggered by the reviewer's comment, we have now also calculated the theoretical conductance in a much greater magnetic field range considering the simple case of a single subgap state at energy E_0 (close to the gap edge Δ_{ind}) dispersing linearly in magnetic field due to Zeeman splitting. The superconducting gap edge of a proximitized nanowire is essentially comprised of a collection of Andreev bound states. Applying a field causes these states to peel off from the gap edge and disperse with a unique g factor associated with each of these states. Although our experimental observations suggest the presence of a few subgap states with different g factors, our numerical simulation already captures well the most relevant transport features such as the splittings at $V_{SD} \sim 2\Delta_{ind}/e$ and $\sim \Delta_{ind}/e$. For clarity, we have also included schematics of the involved first order transport processes for increasing magnetic field. These plots have been included in the Supplementary Information as the new Fig. S19. The caption of this figure is:

“Multiple Andreev reflections in the presence of a subgap state. **a** Calculated conductance in the presence of a subgap state at energy E_0 . Both quantities Δ_{ind} and E_0 vary with magnetic field. **b-d** Schematics of the first-order multiple Andreev reflection processes for different magnetic fields increasing from the bottom to the top panel. The superconducting gap is varied accordingly.”

In Supplementary Section III.B, we have modified the original text by removing the sentence “Nevertheless, the low-energy transport ... our model.” and adding this paragraph:

“To better understand the transport features in Fig. 3c, we simulated the conductance assuming a single subgap state whose energy evolves linearly in the magnetic field as $E_0 = \pm(E_{B_{\parallel}=0} - \frac{1}{2}g\mu_B B_{\parallel})$, where $g = 18$ and $E_{B_{\parallel}=0} = 210 \mu\text{eV}$. The result is shown in Fig. S19a, where we have assumed a junction transmission of $T_1 = 0.065$ and a magnetic field dependence of the gap given by $\Delta_{ind} = \Delta_0(1 - B_{\parallel}^2/B_c^2)$ [Douglass1961, Morris1961] with $B_c = 1.1$ T and $\Delta_0 = 236 \mu\text{eV}$. In Figs. S19b-d, we illustrate the quasiparticle transport processes for different magnetic fields. The conductance peaks at $V_{SD} = \pm 2\Delta_{ind}/e$ correspond to an energy difference of $2\Delta_{ind}$ as denoted by the red arrows. If the electron transfer involves a subgap state at energy E_0 on one side of the junction, the corresponding bias voltage is $V_{SD} = (\Delta_{ind} + E_0)/e$ (Fig. S19d). As the magnetic field is increased, the subgap state moves

to lower energies. Once the state is at zero energy ($E_0 = 0$), electrons only require an energy of $eV_{SD} = \Delta_{ind}$ (Fig. S19c). In Fig. 3c of the main text, this occurs around $B_{||} = 0.4$ T. As the subgap state crosses zero energy, electrons again require an energy of $eV_{SD} = \Delta_{ind} + E_0$ to cross the junction via this state (Fig. S19b). If the junction is more transmissive, as is the case for Fig. 3c of the main text, also a MAR process occurs, identified by the conductance peak that emerges at $V_{SD} \sim 0.24$ mV at zero field and is associated with an energy Δ_{ind} . In addition, when the subgap state moves to lower energies due to the Zeeman effect, it also allows for a MAR process to occur, which results in a splitting of the MAR peak. With increasing magnetic field, the superconducting gap on both sides of the junction shrinks, resulting in the scenario shown in Fig. S19a, where the subgap state moves as a function of magnetic field from $V_{SD} = 2\Delta_{ind}/e$ down to Δ_{ind}/e and back up to $2\Delta_{ind}/e$. Considering that multiple subgap states can peel off from the gap edge with different associated g factors, a rich and complex pattern can occur. This interpretation of the involved transport processes is supported by the numerical simulation.”

8) For devices shown in Figs. 4a and 5a, again, it is good to add description about how are Ti/Au contacts are made and what are their thickness. What are the impact of this fabrication procedure on the device part made by shallow-wall technique?

Details of the fabrication of the leads are already discussed in Supplementary Section I.C “Additional Fabrication Steps for N-S Devices”. Ti/Au contacts are made via electron-gun deposition of 10 nm of Ti (wetting layer) and 120 nm of Au (this number has been corrected in the revised manuscript) at a pressure of around $8 \cdot 10^{-8}$ mbar. Prior to the metal deposition, Ar^+ Kaufmann sputtering was performed to remove the native oxide in the contact region. The normal leads are patterned via standard lift-off technique. Resist baking that is conventionally used to remove solvents from the resist after spin-coating is avoided by using vacuum pumping instead, in order to keep this single lithography step minimally invasive.

Accordingly, we have modified the following sentence in Supplementary Section I.C:

“It consists of EBL patterning (**solvents are removed from the resist via vacuum pumping instead of conventional resist baking to accommodate the low thermal budget**), 40 s of argon ion milling at $1.5 \cdot 10^{-3}$ mbar with a commercial Kaufmann source in the load lock of an electron-beam evaporator, and in-situ evaporation of 10 nm/**120 nm** of Ti/Au **at a pressure of $8 \cdot 10^{-8}$ mbar** followed by lift-off in acetone.”

Regarding the impact of the N contact fabrication on the hybrid nanowire segment see also our answer to question 4 of reviewer 1. The devices presented here require at most one post-interface fabrication step, which is used for device types with normal metal leads (gates and dielectric are already incorporated in the substrate). This single step is only minimally invasive since the hybrid InSb/Al segments remain untouched. The fact that the N-S devices that we studied do not show any subgap states at zero magnetic field and that the induced gaps are hard and well described by the BTK theory, with a similar magnitude of the induced gaps as for the Josephson junctions that did not require any post-interface fabrication, demonstrates that this single metal deposition step outside of the hybrid segment did not negatively impact the device performances.

Moreover, in Section II.D, we introduce two alternative solutions offered by the shadow-wall technique to entirely omit any post-interface fabrication step:

“Alternatively, Al leads that are defined by the shadow walls – microns away from the N-S junction – can serve as N contacts but require additional bottom gates to render all nanowire segments fully

conducting (cf. Fig. 1b). Another option to fabricate N contacts in situ involves using two deposition angles, which we describe in detail elsewhere [Borsoi2020].”

9) Note a misprint in caption of Fig. 4c, i.e., (a) should be (b). Note also a misprint in Fig. 4e, i.e., $650 \mu\text{eV}$ should read $650 \mu\text{V}$.

We thank the reviewer for noticing these typos. We have adjusted the caption of Fig. 5 accordingly.

10) In Section D, it states that at low V_{tg} , subgap conductance is suppressed by about two orders of magnitude when compared to normal-state conductance. Is this enough to claim achieving a hard gap in the hybrid nanowires?

Transport at N-S hybrid junctions with hard induced superconducting gaps obeys the Blonder-Tinkham-Klapwijk (BTK) theory presented in Blonder et al., Phys. Rev. B 25, 1982. Therefore, the excellent match between our data and the prediction of the BTK theory illustrated in Figs. S23 and S25 is per se enough to claim the hardness of the gap. Moreover, we show in Fig. 4 that the zero-bias conductance as a function of the normal-state conductance follows the Beenakker formula without any fitting parameter reasonably well across 4 orders of magnitude. The Beenakker formula is based on the BTK theory and captures the hardening of the induced gap in the weak-tunnelling limit at zero field and for sufficiently large chemical potential (see Liu et al., Phys. Rev. B 96, 2017).

With this in mind, the suppression of the subgap conductance by about two orders of magnitude compared to the normal-state conductance in the tunnelling regime is redundant, but we consider it relevant information to allow the reader to draw comparisons with previous works such as Krogstrup et al., Nat. Mater. 14, 2015 and Gazibegovic et al., Nature 548, 2017.

11) In Section E, to my understanding, no convincing evidence on the correlation between ZBPs at the two ends has been provided. Thus, I would not say that the title of the section is appropriate here. Could you consider a revision?

We agree with the referee’s assessment that the measurement in Fig. 5 alone is not sufficient to claim correlation between the ZBPs. While the super-gate dependence in Figs. S28 and S29 shows simultaneous appearance of these ZBPs at both ends of the device, a systematic study of the ZBP phase diagram as a function of magnetic field and super gate is required to find compelling evidence for correlation. For this reason, we have changed the title of this section to “Emergence of Zero-Bias Peaks at Both Nanowire Ends”.

12) In Fig. 6b, I would not think that this device will be a successful three-terminal device, since the SC loop in the device is hard to be bounded (or am I wrong?). The authors should commend on this or replaced the device picture with a more proper one.

Fig. 6b shows the implementation of a ‘loop qubit’ device using the shadow-wall technique. A three-terminal device is presented in Fig. 5a. These are two rather different types of devices, but both exploit the connection of the Al from the nanowire to the substrate (see EDX image in Fig. 2c). In the ‘loop qubit’ device, the Al loop connects one hybrid segment to the other (see schematic in Fig. 15b of Karzig et al., Phys. Rev. B 95, 2017) and it is critical that this loop is at a floating potential (it will *not* be bonded and it should *not* be connected to ground). In contrast, in the three-terminal device the central Al lead extends to a bond pad and it functions as an electrically grounded drain. We are confident that the current text in Section III, together with the citation

of Karzig et al., Phys. Rev. B 95, 2017, already clearly conveys this aspect.

13) Some typeset issues: A more important one is that please always use a “-“ in the phrase like “normal metal-superconductor”, or always use a “/” in such phrases.

We have corrected this phrase and changed the expression to “normal metal/superconductor”.

Reviewer #3 (Remarks to the Author):

The manuscript presents a novel, unconventional technique to fabricate hybrid semiconductor-superconductor nanowire devices without the need for resist coating and etching processes that could degrade the device surface. The technique, based on aluminum angle evaporation through predefined shadow walls, is original and well characterized by means of multiple inspection techniques based on SEM and TEM. An abundance of illustrative images is provided both in the main text and in the supplementary material.

In order to validate their fabrication technique, the authors have performed low-temperature transport experiments on a set of two-terminal and three-terminal hybrid devices. Fig. 3, as well as Figs. S5 & S6, show data for Al-InSb nanowire-Al devices with a relatively short ‘channel’ length of 110-150 nm, comparable to the nanowire diameter.

We thank the referee for the detailed review of our work and for providing critical suggestions.

1) a) Superconductor-semiconductor junctions can show relatively high transparency, validating the surface cleaning process by atomic hydrogen radical cleaning. From the fitting of MAR features, the authors extract transparencies close to unity for the first subband. In Fig. S5, conductance quantization is claimed at large finite bias, i.e. well above the superconducting gap ($V_{SD} = 10 \text{ mV} \gg \Delta$). This claim is questionable though: the steps are barely visible (which is quite typical for nanowire devices); to my understanding, the series resistance cannot be independently measured.

Indeed, Fig. S8b (previously Fig. S5b) is not a very compelling demonstration of quantization but (as pointed out by the referee) this is quite typical for nanowire devices. Only two conductance plateaus might be identified at integer multiples of G_0 , which is a first basic signature for ballistic transport at zero magnetic field. Additional demonstration of ballistic transport of the same batch of nanowires at finite magnetic field is provided by Badawy et al., Nano Lett. 19, 2019.

We have softened the claim of ballistic transport in Fig. S8 in Supplementary Section II:

“The current and differential conductance in the normal state ($V_{SD} = 10 \text{ mV}$) display a steplike increase as a function of V_{BG} (Figs. S8a,b). **The first two steps approximately align with the quantized values expected for one-dimensional transport, providing possible hints for ballistic transport at zero magnetic field.**”

We have clarified in the caption of Fig. S8b that we in fact only subtract the series resistance of the setup (line and filter resistances as well as the input resistance of the pre-amplifier) without considering the entirely unknown (but possibly very small) contribution of contact resistances: “... after subtracting the series resistance **of the setup** ...”. We have recognized a small error in the subtracted R_s for this panel, which is now corrected.

2) Fig. 4 shows data for a normal metal-InSb nanowire-Al device. Suppressed sub-gap conductance is shown in the low-transparency regime. Upon increasing the voltage on tunnel gate (hence lowering the tunnel barrier), a step-like increase of the sub-gap conductance is observed. The above-gap conductance exhibits a quasi-plateau around the conductance quantum. Simultaneously, the zero-bias conductance shows a peak at about twice the conductance quantum, followed by a pronounced dip on the right side of Fig. 4d. Fig. 4 is used to claim ballistic transport in the presence of Andreev reflection, but I have several concerns about that:

2) a) Based on the discussion in Supplementary Section IV, data in Fig. 4 are plotted following the subtraction of a series resistance. Surprisingly, I could not find that mentioned anywhere in the main text, and the value for the subtracted series resistance was not specified. To claim $G=2e^2/h$, this point is crucial.

We thank the referee for highlighting this important aspect. To claim quantization of conductance as evidence for ballistic transport, knowledge of the exact series resistance of the setup is indeed critical. We have taken into account all contributions to the series resistance, including line resistances, filter resistances as well as the input resistance of the pre-amplifier. For all datasets, this setup-dependent series resistances has been subtracted from the raw data, which is of course mandatory for all transport measurements, without making any assumptions about a possible contact resistance of the interface between the metal leads and the InSb channel. We know that the contact resistance is very small compared to the series resistances of the setup, but the precise magnitude of this contribution is inherently unknown in a two-terminal measurement geometry. The values of the subtracted series resistance are setup-dependent and bear no significance for the device physics. They are disclosed together with the raw measurement data and the analysis code in the data repository at <https://doi.org/10.5281/zenodo.3954465>.

In the revised version of the manuscript, this aspect is clarified in the Methods section by adding the following sentence:

“For all two-terminal conductance measurements we only subtract setup-related series resistances without making any assumptions about additional contact resistances of the metal-semiconductor interface.”

2) b) To access the normal-type regime, the conductance is measured at relatively high bias voltage, where the effects of disorder-induced scattering and possible Coulomb-blockade effects are largely washed out (btw, the same is true for the data of Figs. S5 & S6). Not surprisingly, the low-bias conductance shows much stronger modulations.

Scattering due to residual disorder cannot be fully excluded but we expect that those effects would persist also far beyond the gap edge. The normal-state conductance, G_N , in Figs. 4d,e is extracted at a bias voltage $|V_{SD}|$ of around 0.6 mV. Taking the linecut closer to the gap edge would not represent the actual value of G_N since the conductance is enhanced by the proximity to the superconducting coherence peak. The subgap conductance as a function of the normal-state conductance is well described by the Beenakker model over several orders of magnitude without any fitting parameter (Fig. 4e). This is strong evidence for the fact that the subgap conductance is dominated by Andreev processes. If disorder-induced scattering would influence only the subgap conductance this would be visible as outliers in Fig. 4e, but this is not the case.

Localization effects due to Coulomb blockade can also be ruled out. Those would manifest as diamond-like features crossing the gap, visible both in the subgap conductance as well as the above-gap conductance.

However, the conductance map in Fig. 4b clearly shows that this is not the case. The low bias conductance, even for very clean quantum point contacts, can be affected by phase coherent effects that are averaged out at larger bias voltage (such as Fabry-Pérot resonances, see Kretinin et al., Nano Lett. 10, 2010), but these features would not constitute an argument against the presence of ballistic transport.

We agree with the referee, that there are several jumps in the conductance data, which we attribute to the trapping and detrapping of electrons in the gate dielectric, which are reproducibly activated at certain energies. One of these charge jumps occurs at $V_{TG} = 0.65$ V and it is visible in both G_S and G_N (i.e., over the entire bias-voltage range). Looking at the blue linecut in Fig. 4c at a large tunnel-gate voltage, we can see that the Andreev plateau is stable across the entire bias range of $\pm\Delta_{ind}/e$ and it is located right at the value of $4e^2/h$ expected for unity transmission, a feature that (to our knowledge) has not been observed in the nanowire community before (cf. Zhang et al., Nat. Commun. 8, 2017, where G_S reaches only up to $\sim 3e^2/h$). Zero-bias Andreev enhancement is even more sensitive to disorder in the junction than the normal-state conductance, and the presence of this pronounced Andreev plateau is compelling evidence for the excellent junction quality.

2) c) a strong conductance dip is observed for $V_{TG} \sim 0.8$ V. The explanation in terms of “inter-subband scattering” is not convincing. To begin with, the authors refer to the onset of conduction through the second subband, but this cannot be seen from the data shown.

The onset of conductance via the second subband is likely related to the rise in conductance around $V_{TG} \sim 0.85$ V. The strong suppression of G_S around $V_{TG} = 0.8$ V is attributed to inter-subband mixing due to residual disorder, i.e. redistribution of the transmission amplitudes (T_i) among available subbands in the proximitized nanowire such that $G_S = \sum_i \frac{4e^2}{h} \frac{T_i^2}{(2-T_i)^2}$ is less than $G_N = \sum_i \frac{2e^2}{h} T_i$. This phenomenon and the analogous conductance behaviour have been measured and explained in a previous study (see e.g. Supplementary Fig. 4 of Zhang et al., Nat. Commun. 8, 2017) and also observed in other experiments (e.g., Gül et al., Nat. Nanotechnol. 13, 2018 and Chang et al., Nat. Nanotechnol. 10, 2015). Nevertheless, since we do not focus on this particular phenomenon in detail in the current work, we have softened the claim in the revised manuscript and pointed out the possible role of residual disorder:

“While the subgap conductance reaches up to $2G_0$, it drops again ~~before the chemical potential reaches the bottom of the second confinement subband at $V_{TG} \sim 0.8$ V, possibly~~ due to inter-subband scattering **as a result of residual disorder [Chang2015, Heedt2016, Zhang2017, Gül2018].**”

3) Fig. 5 shows data for a three-terminal device (one superconducting contact covering a 1-um-long segment of the nanowire, and two normal-type contacts at the edges). Around 1 T, zero-bias conductance peaks develop at both edges with no apparent correlation. A second data set is shown in Fig. S24. The peak heights observed at both edges are relatively high, possibly close to $2e^2/h$ (once again, I wasn't able to find any value of the subtracted series resistance. The authors only refer to an uncertainty of 0.5 kOhm on the series resistance, but how can they say that?).

The local conductance measured simultaneously at the two opposite N-S junctions of the three-terminal device in Fig. 5 reveals zero-bias peaks that emerge at similar magnetic field values. The onset field is 0.85 T at the left junction and 0.78 T on the right side (cf. our response to question 5 of referee 1). In Fig. S28, at a different super-gate voltage, the two ZBPs appear at approximately the same magnetic field and show a similar response to a variation in the super-gate voltage. However, it is indeed possible that this correlation is accidental and

that the two ZBPs arise from localized Andreev bound states. It is possible that the ZBPs in Fig. 5 are signatures of localized states that accidentally appear at similar magnetic fields and super-gate voltages. To prove whether the ZBPs in this device can be attributed to Majorana zero modes, it is imperative to demonstrate a closing and reopening of the induced gap in the nonlocal conductance before the correlated appearance of ZBPs at larger magnetic field. For this reason, the results in Fig. 5 should be viewed as a proof-of-principle demonstration of a local correlation measurement. Experimentally, this is a significant step forward and marks the starting point for the systematic search for a topological phase transition by combining the correlation (and robustness) of local ZBPs with the evolution of the induced gap in the nonlocal conductance. This evolution of the induced gap is currently being studied via nonlocal conductance measurements between the nanowire ends and is beyond the scope of the current manuscript (the analysis of these very large datasets calls for a study of its own).

As mentioned in our response to the previous question, we did not subtract any additional contact resistances but only setup-related series resistances that necessarily must be considered in every transport measurement. To reflect a possible uncertainty in this procedure, we have made a conservative estimate for possible additional contact resistances. This conservative estimate ($0.5 \text{ k}\Omega$) is meant to give an idea of the largest possible uncertainty in the conductance data.

4) Overall, based on the results of the different transport experiments, I do not see a clear benefit from the presented fabrication technique over previously used approaches. Owing to the avoided device processing, I would have expected some clear improvement in the transport properties of the nanowires, e.g. a clearer evidence of ballistic transport, but what is shown does not go beyond the state-of-the-art. The same group and other groups have reported results of comparable level, e.g. Zhang et al., *Nature* 556, 74–79(2018); Gül et al, *Nature Nanotechnology*, 13, 192–197(2018); Abay et al. *Nano Letters*, 13, 3614 (2013); Xiang et al. *Nature Nanotechnology* volume 1, 208–213(2006). For this reason, I would not recommend publication in *Nature Communications*.

We thank the referee for addressing this important point. In our manuscript, we exploit atomic hydrogen cleaning treatment before contacting the InSb nanowires with Al thin films. As correctly pointed out, this is similar to Gazibegovic et al., *Nature* 548, 2017 and Zhang et al., *Nature* 556, 2018. As discussed in our response to question 1 of referee 1, shadow-wall lithography offers pristine interface quality on par with state-of-the-art in-situ deposition methods, but it effectively removes the variations inherent to other methods and introduces unprecedented flexibility and fundamentally new and more advanced device designs. The pre-patterned shadow walls allow for substantially simplified fabrication, while retaining state-of-the-art electronic transport properties. Signatures of ballistic transport are limited by the crystalline quality of the underlying semiconductor. The nanowires employed here are identical to those reported by Badawy et al., *Nano Lett.* 19, 2019 and major improvements of the normal transport signatures significantly beyond the best results in the literature cannot be expected. Instead, the milestone reached in this paper is the scalable fabrication of hard-gap super-/semiconductor hybrids, which enables important new device types and does not require in-situ processing. It does not rely on customized designs but allows for convenient blind fabrication.

One of the critical challenges in the field is to probe the energy spectrum at both ends of a hybrid nanowire simultaneously. To the best of our knowledge, shadow-wall lithography introduced in this manuscript is the only method that allows to create the required three-terminal device geometry with a grounded magnetic-field resilient superconductor without etching. The electrical thin-film connection from the substrate to the shell is a critical feature that none of the other shadow-deposition methods can offer. Conventional techniques require

to selectively etch away the oxide of the Al shell, deteriorating the integrity of the thin film, and/or to etch away the superconducting shell itself, which causes disorder at the InSb surface. Thanks to the shallow-angle deposition, the superconducting Al lead is very thin and thus magnetic-field resilient. Otherwise, the gap of the parent superconductor would close before the induced gap in the nanowire, preventing us from detecting a topological phase transition in the bulk of the hybrid (see Rosdahl et al., Phys. Rev. B 97, 2018).

Among the references mentioned by the referee only Zhang et al. have demonstrated a hard gap, representative for other works that have realized hard induced gaps via shadow-deposition methods. However, the device yield is typically limited by the involved fabrication steps and by ageing of the superconductor/semiconductor interface. As stated by Zhang et al., arXiv:2101.11456, 2021 (appendix A), conventional shadow deposition resulted in 11 out of 60 devices showing “good transport characteristics” and having functional gates. With our technique, the average Josephson junction yield is typically around 80% by fully eliminating post-processing steps (see Supplementary Section I.F). As a bonus feature, all nanowires are aligned along the same direction, which simplifies the alignment of the magnetic field – a key requirement to form a topological phase in these hybrid nanowires. Moreover, our method is not constrained by the low thermal budget of the InSb/Al hybrid (chapter 5 of S. Gazibegovic, Ph.D. thesis, Eindhoven University of Technology, 2019) and can incorporate high-temperature dielectric deposition. This allows us to employ high-quality Al_2O_3 that is deposited via atomic layer deposition at 300°C before creating the delicate semiconductor-superconductor interface. Previous works, including those mentioned by the referee, suffer from gate instabilities caused by room-temperature sputtered dielectrics, as discussed by Zhang et al., arXiv:2101.11456, 2021 (appendix D). A major improvement in transport properties to the state-of-the-art is the particularly clean N-S data with a compelling demonstration of Andreev enhancement. Another benchmark for the significant improvement in hybrid device quality is the significant increase in $I_{sw}R_N/\Delta_{ind}$ compared with previous InSb-based Josephson junctions (see our response to question 2 of referee 1).

Our technique represents an enabling technology that provides various new opportunities. It is a platform to conveniently try out new material combinations with high yield and without changing the fabrication concepts. In particular, it is crucial for Majorana research and enables basic qubit architectures (see Fig. 6), but it can be applied well beyond that field for other quantum devices, as exemplified by the SQUID in Fig. S30 that also requires connections via the substrate, which are beyond the capabilities of previously developed techniques.

List of additional changes in the bibliography:

- We have added a new reference to Pan et al., Phys. Rev. B 103, 2021, covering local and nonlocal conductance measurements in three-terminal Majorana nanowires.
- The reference to the preprint by Yu et al., arXiv:2004.08583, 2020 has been updated with the peer-reviewed version Yu et al., Nat. Phys. 17, 2021.

REVIEWERS' COMMENTS

Reviewer #1 (Remarks to the Author):

The authors have provided satisfactory answers to all my questions and comments. I recommend publication without further changes.

Reviewer #3 (Remarks to the Author):

The authors have largely improved the original version of the manuscript and have provided satisfactory answers to my criticisms, and to the requests from the other reviewers. Relaxing certain unnecessary claims makes this work more solid and suitable for publication in Nature Comm. It will be interesting to see if the proposed fabrication technique will eventually be adopted by the community.

Response to the reviewers' comments:

We thank the referees for reviewing the revised version of our manuscript and for the positive feedback towards its publication. Below, we list the remarks from the referees in black and our response in blue.

Reviewer #1 (Remarks to the Author):

The authors have provided satisfactory answers to all my questions and comments. I recommend publication without further changes.

Thank you for reviewing our revised manuscript. We very much appreciate your valuable feedback during the first round of review, which has helped improve the overall quality and clarity of our paper.

Reviewer #3 (Remarks to the Author):

The authors have largely improved the original version of the manuscript and have provided satisfactory answers to my criticisms, and to the requests from the other reviewers. Relaxing certain unnecessary claims makes this work more solid and suitable for publication in Nature Comm. It will be interesting to see if the proposed fabrication technique will eventually be adopted by the community.

Thank you for reviewing the revised version of our paper and for your commendatory words. We agree that the revision made the manuscript more solid and that it has significantly improved in clarity.

Changes in the final submission:

- For improved readability, we have changed the first sentence of the abstract to
“The realization of hybrid superconductor–semiconductor quantum devices, in particular a topological qubit, calls for advanced techniques to readily and reproducibly engineer induced superconductivity in semiconductor nanowires.”
- On the first page, we replaced reference 19 (Mourik et al. Science 336, 2012) with Lai et al. Phys Rev. B 100, 2019 and Pan et al. Phys. Rev. B 103, 2021, as these are better suited in this context. All three references remain part of the bibliography, only the order has changed.
- On page 6, we have corrected the condition for the maximal potential inhomogeneity that allows for a continuous topological phase:
“A topological phase with well-separated MBS requires that potential inhomogeneities along the hybrid segment, $\Delta\mu$, are much smaller than the **width of the topological phase**, $2\sqrt{E_Z^2 - \Delta_{\text{ind}}^2}$, where E_Z is the Zeeman energy [1,55].”
- We have made several other minor changes and additions, which are highlighted in orange in the manuscript.